# EvoPrompting: Language Models for Code-Level Neural Architecture Search

**Angelica Chen**[*]
New York University
angelica.chen@nyu.edu

**David M. Dohan**[†]
OpenAI
david@ddohan.com

**David R. So**[†]
Jane Street
david.r.so.ai@gmail.com

## Abstract

Given the recent impressive accomplishments of language models (LMs) for code generation, we explore the use of LMs as general adaptive mutation and crossover operators for an evolutionary neural architecture search (NAS) algorithm. While NAS still proves too difficult a task for LMs to succeed at solely through prompting, we find that the combination of evolutionary prompt engineering with soft prompt-tuning, a method we term EVOPROMPTING, consistently finds diverse and high performing models. We first demonstrate that EVOPROMPTING is effective on the computationally efficient MNIST-1D dataset, where EVOPROMPTING produces convolutional architecture variants that outperform both those designed by human experts and naive few-shot prompting in terms of accuracy and model size. We then apply our method to searching for graph neural networks on the CLRS Algorithmic Reasoning Benchmark, where EVOPROMPTING is able to design *novel* architectures that outperform current state-of-the-art models on 21 out of 30 algorithmic reasoning tasks while maintaining similar model size. EVOPROMPTING is successful at designing accurate and efficient neural network architectures across a variety of machine learning tasks, while also being general enough for easy adaptation to other tasks beyond neural network design.

## 1 Introduction

Scaling of Transformers (Vaswani et al., 2017) has produced language models (LM) with impressive performance. Beyond achieving state-of-the-art results on conventional natural language processing tasks, these LMs demonstrate breakthrough technical capabilities, such as learning how to code (Chen et al., 2021), doing math (Noorbakhsh et al., 2021), and solving reasoning problems (Wei et al., 2022). Yet, despite these strides, several works have noted LMs' current limitations in solving complex problems and creating novel solutions (Qian et al., 2022; Dakhel et al., 2022). In this work, we improve upon a base LM's ability to propose novel and diverse solutions to complex reasoning problems by iteratively evolving in-context prompts and prompt-tuning the LM. We call this technique EVOPROMPTING and demonstrate its success on the difficult task of deep learning architecture design. Our key finding is that, while LMs perform poorly at designing novel and effective neural architectures via naive few-shot prompting, EVOPROMPTING enables LMs to create novel and effective deep neural architectures, particularly when combined with prompt-tuning methods.

---

[*]Work done while a Student Researcher at Google DeepMind.
[†]Work done while at Google DeepMind.

37th Conference on Neural Information Processing Systems (NeurIPS 2023).

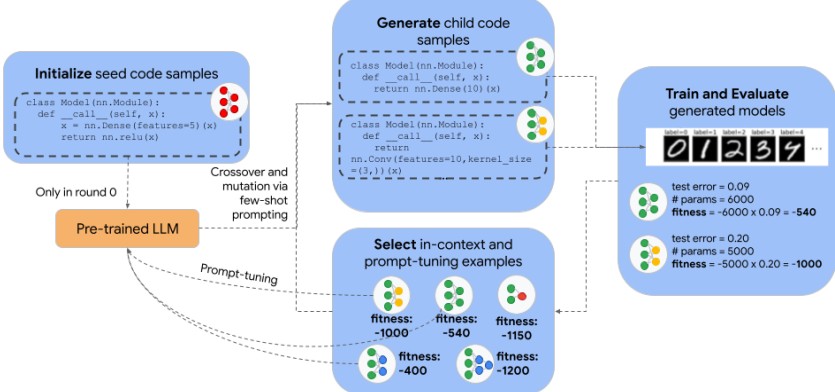

Figure 1: An overview of EVOPROMPTING. After *initializing* the search with a handful of manually designed program seeds, the meta-learning loop begins. First, our code-pretrained LM uses the seeds as in-context prompt examples to *generate* candidate architectures. Those candidate architectures are then *trained* on the task training data and *evaluated* on the task validation set. Next, the most fit members of the population are *selected* as in-context examples for the next meta-learning loop and all evaluated individuals are used as training data for prompt-tuning the LM. From there, the meta-learning loop begins again.

EVOPROMPTING is based on the recently popularized practice of in-context prompting. Prompting is the technique of conditioning a LM's decoded output on a custom prefix known as a *prompt*, which can include natural language task instructions or a few input-output examples. The prompt is used only at inference time and requires no gradient updates (Brown et al., 2020). In past work, prompting has been demonstrated to elicit impressive performance on a wide variety of tasks without requiring task-specific fine-tuning (Sanh et al., 2021; Wei et al., 2022; Kojima et al., 2022). Here, we leverage LM prompting for the task of designing improved deep learning architectures.

To engineer adequately powerful prompts, we draw inspiration from existing ideas in the field of neural architecture search. There, evolution has long been used to search over discrete spaces to efficiently discover improved deep learning architectures (Yao, 1999; Real et al., 2017). However, evolutionary approaches typically require careful manual design of a discrete search space (*e.g.* a small set of known convolutional neural network components, as in Real et al. (2017) or TensorFlow primitives, as in So et al. (2021)). As a result, the performance of the evolutionary algorithm is then sensitive to and possibly limited by the design of the search space. In EVOPROMPTING the LM's vocabulary replaces the search space, which both increases the flexibility of the search and reduces reliance on manual design. The LM is also an *adaptive* mutation/crossover operator, in the sense that it can be improved round over round via prompt-tuning. Furthermore, EVOPROMPTING also improves on naive few-shot prompting by using an evolutionary search approach to iteratively improve the in-context examples for few-shot prompting.

To demonstrate the effectiveness of this method, we first do extensive testing and analyses on the relatively low-compute problem of MNIST-1D (Greydanus, 2020). The key finding of these experiments is that EVOPROMPTING is capable of producing conventional convolutional architectures superior to published manually designed models (Section 4.1). In Section 4.2 we then apply our method to the more challenging task of designing graph neural networks using problems from the CLRS Algorithmic Reasoning Benchmark (Veličković et al., 2022), where EVOPROMPTING generates novel architectures that outperform state-of-the-art models on 21 out of 30 algorithmic reasoning tasks (Appendix 3).

The contributions of this work are summarized as follows:

1. We propose EVOPROMPTING, a method that utilizes evolutionary search to create and curate data to improve LM in-context prompting examples. Although this work focuses on the specific task of neural architecture design to develop this method, EVOPROMPTING is generally applicable to LM tasks that rely on in-context learning (ICL) or prompt-tuning.

2. A study applying LMs to code-level neural architecture design. Our experiments demonstrate that applying few-shot prompting alone to neural architecture design is unsuccessful, but few-

shot prompting with EVOPROMPTING enables LMs to create architectures that outperform those designed by human experts.

3. Novel graph neural network architectures that were discovered using EVOPROMPTING. These architectures outperform the current state-of-the-art architecture, Triplet-GMPNN (Ibarz et al., 2022), on 21 out of 30 CLRS Algorithmic Reasoning Benchmark tasks (Appx. 3).

## 2 Related Work

**LMs for code generation**  Scaling Transformers (Vaswani et al., 2017) is currently a popular route for reliably creating state-of-the-art natural language systems (Brown et al., 2020; Du et al., 2021; BigScience Workshop et al., 2022; Zhang et al., 2022; Thoppilan et al., 2022; Chowdhery et al., 2022). Many works have observed that large LMs are capable of performing technical tasks such as writing code (Chen et al., 2021), doing math (Noorbakhsh et al., 2021), and solving complex reasoning problems (Wei et al., 2022). Our work is most closely related to efforts that have applied LMs to coding tasks (Chen et al., 2021; Odena et al., 2021; Xu et al., 2022; Wang et al., 2021; Ahmad et al., 2021; Feng et al., 2020), since our technique proposes architectures in code.

**Prompting**  Brown et al. (2020) demonstrated that LMs can be prompted with in-context examples to steer LM decoding towards solving problems in-context without gradient updates. Numerous works have utilized this prompting to further boost LM abilities (Sanh et al., 2021; Wei et al., 2022; Kojima et al., 2022). Others have focused on optimizing these prompts  (Min et al., 2022; Liu et al., 2021) as via approaches such as augmentation with retrieval systems  (Rubin et al., 2021), permutations of few-shot examples (Lu et al., 2021; Zhao et al., 2021), generating prompts via LMs (Zhou et al., 2022), and instruction-tuning (Wei et al., 2021; Ouyang et al., 2022; Sanh et al., 2021). From the perspective of Dohan et al. (2022), prompts are parameters that can be tuned using probabilistic inference techniques. Brooks et al. (2022) proposes using few-shot prompts to implement both the rollout policy and world model of a policy iteration algorithm. Our EVOPROMPTING method extends these efforts by proposing evolutionary search as a means to both better design prompts for ICL and tune the base LM to use the prompt more effectively.

**Evolutionary Algorithms**  Our method is closely related to evolutionary neural architecture search (NAS) (Real et al., 2017, 2018; Elsken et al., 2018; So et al., 2019; Liu et al., 2020), in which architectures are represented as discrete DNAs, and evolved and filtered based on fitness metrics that assess architecture performance.  However, our method can search over arbitrary strings of code, whereas conventional evolutionary NAS algorithms rely on hand-crafted search spaces that can strongly bias and contrain the search (Li & Talwalkar, 2019; Sciuto et al., 2019; Bender et al., 2020; Real et al., 2020; So et al., 2021). A work close to ours is Lehman et al. (2022), in which an LM is fine-tuned to produce Python code diffs given one of three fixed messages that describe what should be changed, and then used as the mutation operator in an evolutionary algorithm. Their work is validated on the Sodarace domain. Our work differs in that we use an LM as a crossover operator, without specifying the class of changes to make, which may offer greater flexibility. Furthermore, we evaluate our approach on the real-world task of NAS, rely on mixed temperature sampling of the LM for diversity instead of using a QD algorithm, and also use prompt-tuning in our algorithm. We choose not to use a QD algorithm such as MAP-Elites since this approach requires the design and discretization of a descriptor space, which is complex and difficult to hand-design for the space of all possible neural networks.

Another concurrent work is Meyerson et al. (2023), which uses an LM as a crossover operator to produce variations of text-based genotypes in the domains of symbolic regression, text sentiment, images, and Sodaracer programs. Like Lehman et al. (2022), they use MAP-Elites to trade off quality with diversity in two of the domains and demonstrate that their overall algorithm reliably produces a diverse range of outputs. They additionally demonstrated performance comparable to state-of-the-art approaches on the toy task of symbolic regression. Their study varies from ours in a number of ways – we apply our algorithm to the real-world task of NAS, we optimize for a tradeoff between state-of-the-art task performance and model size, we condition on target performance in our prompts, we do not use MAP-Elites, and we use prompt-tuning to iteratively improve the LM's crossover abilities instead.

# 3 EVOPROMPTING Method

## 3.1 Architecture search problem formulation

Let our target task be denoted by $\mathcal{T}$ and $\mathcal{D}$ be a dataset consisting of input-output pairs $(x, y) \in \mathcal{D}$ for task $\mathcal{T}$. Define the probability distribution $\pi_\theta : \mathcal{V} \to \{0, 1\}$ over vocabulary $\mathcal{V}$ as a language/code model parameterized by $\theta$, from which we can sample code segments $c \in \mathcal{V}^*$ (for $\mathcal{V}^*$ the Kleene closure of $\mathcal{V}$, *i.e.* the set of all concatenations of symbols in $\mathcal{V}$). We also have an evaluation function $\text{EVAL}_\mathcal{T}(c, \mathcal{D}) : \mathcal{V}^* \times \mathcal{D} \to \mathbb{R}$ that trains the model architecture given by code $c$ on $\mathcal{D}$ and outputs some real-valued fitness score $s \in \mathbb{R}$, which can be a function of model accuracy and other model characteristics. Our ultimate goal is to identify some set of code samples $c \sim \mathcal{V}^*$ that define neural network architectures that, when trained on $\mathcal{D}$, maximize the reward $\text{EVAL}_\mathcal{T}(c, \mathcal{D})$.

## 3.2 LMs for evolutionary crossover and mutation

The goal of our algorithm is to generate a set $C$ consisting of $k$ neural network architectures that maximize the reward $\text{EVAL}_\mathcal{T}(c, \mathcal{D})$ for arbitrary pairs of $(\mathcal{D}, \mathcal{T})$:

$$\arg \max_{\substack{C = \{c \,|\, c \sim \pi_\theta\} \\ |C| = k}} \mathbb{E}_{c \in C} \mathbb{E}_{(x,y) \in \mathcal{D}} \left[ \text{EVAL}_\mathcal{T}(c, \mathcal{D}) \right] \tag{1}$$

Since this optimization problem is generally intractable, we turn to a black-box evolutionary approach for iteratively generating, scoring, and selecting the best neural network architectures. Indeed, evolution has been demonstrated to perform particularly well in this domain because of how sparse high quality solutions tend to be (Real et al., 2017, 2018). Although evolution has been used for architecture search many times before (Real et al., 2017, 2018; Elsken et al., 2018; So et al., 2019), we improve upon this approach by using an LM for crossover and mutation operations.

Using an LM in this manner has multiple appealing properties. While past evolutionary approaches for neural architecture search have required careful design and specification of a discrete search space (*e.g.* the space of high level modules (Real et al., 2018; So et al., 2019), TensorFlow statements (So et al., 2021), or basic mathematical operations (Real et al., 2020)), our algorithm's search space includes any neural network architecture that can be represented in Python. This allows for greater flexibility and diversity of the output architectures, and reduces the amount of manual design and human bias involved in the algorithm. Furthermore, modern pre-trained LMs are typically trained on massive datasets containing a significant number of source code files. This pre-training process encodes useful knowledge about code structure and functionality that is not otherwise available in evolutionary algorithms. Lastly, LMs can also be used as *self-adaptive crossover operators*, in which the crossover operator is incrementally trained round after round to generate higher reward crossovers.

## 3.3 EVOPROMPTING meta-learning algorithm

Our complete algorithm is described in Algorithm 1. At the core of our algorithm is a scoring function, which describes the general "fitness" of a model on the task at hand. Since higher accuracy can often be achieved simply by increasing the number of parameters in a model, we use the negative product of the validation error and the model size as the fitness (see step 6 in Algorithm 3). More complicated objective functions have previously been used for dual objective neural architecture search (Bender et al., 2020), but we find this simple product works best in our case and requires minimal tuning. Generally the higher the fitness, the better (with some caveats, noted in our description of fitness-based selection below).

The end-to-end meta-learning algorithm has several stages, which we describe below:

**Initialization** We start by setting our global historical population $G$ to the empty list and initializing our current population $P$ with a few seed architectures that are known to be well-designed (step 3 in Algorithm 1), which *warm-starts* the search (So et al., 2019). These seed models are evaluated using the same $\text{EVAL}_\mathcal{T}(c, \mathcal{D})$ function that is used to evaluate new candidate models (see below).

**Algorithm 1** Complete meta-learning evolutionary algorithm using $p_\theta$ as a crossover and mutation operator.

---

1: **Input:** LM $\pi_{\theta_0}$, dataset $\mathcal{D}$, task $\mathcal{T}$, $T$ number of rounds, $m$ number of few-shot prompts per round, $n$ number of samples to generate per prompt, $k$ number of in-context examples per prompt, $p$ number of survivors to select per generation, $\alpha$ the upper threshold for the test error
2: $G \leftarrow []$
3: $P \leftarrow \text{INITIALIZEPOPULATION}(p)$
4: $t \leftarrow 0$
5: **while** $t < T$ **do**
6: $\quad C \leftarrow \text{CROSSMUT}(\pi_{\theta_t}, P, m, k, n)$
7: $\quad C_{\text{EVALED}} \leftarrow \text{FILTERANDEVAL}(C, \mathcal{T}, \mathcal{D}, \alpha)$
8: $\quad G \leftarrow G + C_{\text{EVALED}}$
9: $\quad$ **if** $t < T - 1$ **then**
10: $\quad\quad P \leftarrow \text{GETTOP}(G, p)$
11: $\quad\quad \theta_{t+1} \leftarrow \text{TRAIN}(\theta_t, C_{\text{EVALED}} \setminus P)$
12: $\quad$ **end if**
13: $\quad t \leftarrow t + 1$
14: **end while**
15: Return $\text{GETTOP}(G, p)$

---

**Algorithm 2** The crossover and mutation algorithm, $\text{CROSSMUT}(\pi_{\theta_t}, P, m, k, n)$, where $\text{Uniform}(P)$ denotes the uniform distribution over the set $P$. The set of potential parents $P$ consists of the top examples from the previous round.

---

1: **Input:** LM $\pi_\theta$, population of code samples and fitnesses $P = \{(c, s) \mid c \in \mathcal{V}^*, \text{EVAL}_\mathcal{T}(c, \mathcal{D}) = s\}$, $m$ number of few-shot prompts to create, $k$ number of in-context examples in each prompt, and $n$ number of samples to sample per prompt.
2: $C \leftarrow []$
3: $i \leftarrow 0$
4: **while** $i < m$ **do**
5: $\quad E \leftarrow \{x_j\}_{j=1}^k$, where $x_j \stackrel{i.i.d.}{\sim} \text{Uniform}(P)$
6: $\quad p \leftarrow \text{MAKEFEWSHOTPROMPT}(E)$
7: $\quad C_i \leftarrow \{c_j\}_{j=1}^n$, where $c_j \stackrel{i.i.d.}{\sim} \pi_\theta(\cdot \mid p)$
8: $\quad C \leftarrow C + C_i$
9: $\quad i \leftarrow i + 1$
10: **end while**
11: **Output:** $C$

---

**Algorithm 3** The algorithm for filtering and scoring child models, $\text{FILTERANDEVAL}(C, \mathcal{T}, \mathcal{D}, \alpha)$.

---

1: **Input:** set of code samples $C$, task $\mathcal{T}$, dataset $\mathcal{D}$, evaluation function $\text{EVAL}_\mathcal{T}(c, \mathcal{D})$, upper threshold for error $\alpha$
2: $C_{\text{EVALED}} \leftarrow []$
3: **for** c in $C$ **do**
4: $\quad$ c.error $\leftarrow \text{EVAL}_\mathcal{T}(\text{c}, \mathcal{D})$
5: $\quad$ **if** c.error $< \alpha$ **then**
6: $\quad\quad s \leftarrow -\text{c.model\_size} \times \text{c.error}$
7: $\quad\quad C_{\text{EVALED}} \leftarrow C_{\text{EVALED}} + [(c, s)]$
8: $\quad$ **end if**
9: **end for**
10: **Output:** $C_{\text{EVALED}}$

---

**Crossing over and mutating the parent models**  To mutate and apply crossover to the parents $P$ selected in the last step, we use both the source code and the evaluation metrics of each model in $P$ to create few-shot prompts.

In the last line of the prompt, we create a target set of metrics to condition $\pi_\theta$'s generations on that indicate the desired validation accuracy and model size of the proposed architecture. We set the target

model size as $90\%$ of the minimum model size of the parent models, rounded to the nearest 100 parameters, and the target validation accuracy as $102\%$ of the maximum validation accuracy of the parent models, rounded to the nearest tenth of a percent. We create $m$ such prompts per round, each with $k$ in-context examples selected uniformly at random from $P$. An example of a prompt is shown in Listing 1.

```
1  """Metrics:
2  {'num_params': '4800', 'val_accuracy': '0.865'}
3  """
4  class Model(nn.Module):
5    @nn.compact
6    def __call__(self, x):
7      x = nn.Dense(features=10)(x)
8      return x
9
10   """Metrics:
11  {'num_params': '4300', 'val_accuracy': '0.880'}
12  """
13  class Model(nn.Module):
```

Listing 1: The format of our few-shot prompts. In practice we use 2-shot prompts but we omit the second in-context example here for brevity.

Finally, we use $\pi_\theta$ to generate $n$ samples per prompt, yielding a total of $n \times m$ child samples per round of evolution. We denote this portion of the algorithm as $\text{CROSSMUT}(\pi_{\theta_t}, P, m, k, n)$ (Algorithm 2 and step 6 of Algorithm 1).

**Filtering and scoring child samples**   To score and filter child samples $c$ generated by $\pi_\theta$, we use the evaluation function $\text{EVAL}_\mathcal{T}(c, \mathcal{D})$, which trains the model encoded by $c$ on the dataset $\mathcal{D}$ and returns the lowest validation error encountered during training. All child models are trained for the same number of steps, with the same optimizer hyperparameters. Since our fitness function can potentially be gamed by generating arbitrarily small models, we also add a validation error threshold $\alpha$, which is the upper limit of the validation error that a model can incur without being removed from $G$, the global population. We refer to this function as $\text{FILTERANDEVAL}(C, \mathcal{T}, \mathcal{D}, \alpha)$ (Algorithm 3 and step 7 of Algorithm 1). Lastly, we add the remaining trainable models and their associated fitness scores into $G$ (step 8 of Algorithm 1).

**Fitness-based selection**   After evaluating all child models in the current round, we apply fitness-based selection to identify top candidate models for crossover (step 10 of Algorithm 1). We denote this as $\text{GETTOP}(G, p)$, which refers simply to selecting the $p$ models with the highest fitness scores from $G$. Once these models have been selected, they are permanently removed from the population and cannot be used again as parents for crossover.

**Training** $\pi_{\theta_t}$   Lastly, all child models generated in the current round that were not previously selected for crossover (*i.e.* $C_{\text{EVALED}} \setminus P$) are used to prompt-tune $\pi_\theta$ for the next round (step 11 of Algorithm 1).

## 4   Experiments and Results

We evaluate our meta-learning algorithm on two datasets – MNIST-1D (Greydanus, 2020) and the CLRS algorithmic reasoning benchmark (Veličković et al., 2022). While the former benchmark is lightweight and permits us to do a more thorough analysis of our algorithm, the latter is a newer benchmark that covers 30 different algorithms with more headroom for discovering novel architectures with better performance.

In all of our experiments, our $\pi_{\theta_0}$ (*i.e.* the crossover operator) is a 62B parameter PALM model (Chowdhery et al., 2022) pre-trained on 1.3T tokens of conversational, web, and code documents. It was additionally fine-tuned on a corpus of 64B tokens containing near-deduplicated, permissively-licensed Python source code files from Github. We always sample from $\pi_{\theta_0}$ with

mixed temperature sampling, in which the sampling temperature is selected uniformly from $[0.2, 0.6, 0.8, 1.0]$. Between each round, the model is prompt-tuned (Lester et al., 2021) for 5 epochs with a soft prompt length of 16, batch size of 16, and learning rate of 0.1 (as described in Section 3.3 and Step 11 of Algorithm 1). Unless stated otherwise, we run 10 rounds of evolution with 10 prompts per round and 16 samples generated per prompt, yielding a total of 160 models generated per round and 1600 models generated during the entire search. Duplicate models and un-trainable models are not scored, but do count into the 1600. All other EVOPROMPTING hyperparameters are listed in Appendix A.1.

## 4.1 MNIST-1D

**Dataset** We apply our method first to MNIST-1D (Greydanus, 2020), a one-dimensional, scaled-down version of the MNIST-1D dataset containing examples that are 20 times smaller than the original MNIST dataset. Each example is only 40-dimensional, with 4000 examples in the training dataset and 1000 in test. Since there is no validation dataset, we randomly set aside 500 examples from the training dataset to use as the validation dataset. Despite being more lightweight, MNIST-1D distinguishes more between different architecture types (Greydanus, 2020) than its larger counterpart MNIST (LeCun et al., 1998).

**Meta-learning set-up** Throughout the model search we use the AdamW optimizer (Loshchilov & Hutter, 2019) to train each child model on a single NVIDIA Tesla P100 GPU for 8000 steps, with learning rate 0.01 and batch size 128. We score child models according to the best validation accuracy achieved during training. We also seed the search with 4 seed models - the 3 hand-designed neural baselines from the original MNIST-1D paper (Greydanus, 2020) (GRU, CNN, and MLP) and a fourth, larger CNN model of our own design. All four are implemented with Flax (Heek et al., 2020). We refer the reader to Appendix A.2 for the source code of these seed models.

**Baselines** We compare EVOPROMPTING with the following baselines:

- Naive few-shot prompting: This baseline simply generates code samples $c \sim \pi_{\theta_0}(\cdot|p)$, where $p$ is a 2-shot prompt constructed using in-context examples randomly selected from the seed models (Listing 1). This is essentially an ablation of steps 7-12 in Algorithm 1 with $T = 1$. We increase the number of samples generated per prompt for the naive prompting baseline such that the total number of samples generated by $\pi_\theta$ matches that of the other baselines.

- EVOPROMPTING ( - prompt-tuning): We run the entire algorithm as is, but without prompt-tuning between each round. This is an ablation of step 11 from Algorithm 1

- EVOPROMPTING (random parents): Instead of selecting the most fit models from the last round as parents for the next round, we select parents randomly. This is an ablation of Step 10 in Algorithm 1, which is the GETTOP$(G, p)$ step.

**EVOPROMPTING finds smaller and more accurate models** Figure 2a shows a comparison of the test error and model size of the top 20 models discovered by EVOPROMPTING compared with those of our seed models and three baselines. The points approximate a Pareto frontier, below which each algorithm cannot improve on one dimension without hurting the other. EVOPROMPTING possesses the Pareto frontier closest to the origin, indicating that it finds more optimal models in terms of accuracy and size. In fact, many models in EVOPROMPTING's top 20 discovered models are orders of magnitude smaller than those of the other baselines, while still having lower test error.

We also note that – on this task in particular – EVOPROMPTING excels especially at optimizing convolutional architectures. Many of the top 20 models are narrower and deeper convolutional architectures, with smaller strides, less padding, and no dense layers. These models consistently perform better than the shallower, denser, and wider convolutional architectures seen in earlier rounds of the model search.

Another important aspect of a meta-learning algorithm is the relationship between the number of individuals evaluated and the maximum fitness observed so far, *i.e.* the sample efficiency. Neural architecture search can be an expensive process, with the most open-ended searches requiring the evaluation of trillions of individuals (Real et al., 2020). Thus, it is crucial to identify fit candidates

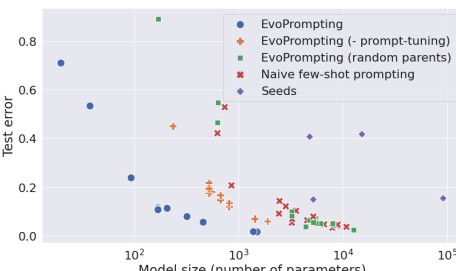

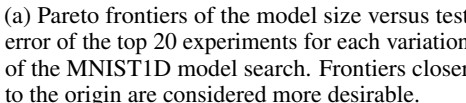

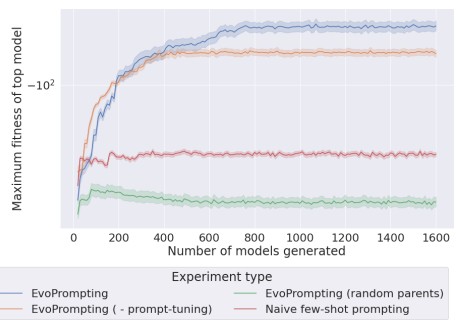

(a) Pareto frontiers of the model size versus test error of the top 20 experiments for each variation of the MNIST1D model search. Frontiers closer to the origin are considered more desirable.

(b) Number of child models generated versus maximum fitness in sample, as estimated using 100 bootstrap samples of size 20 for each point along the x-axis.

Figure 2: EVOPROMPTING discovers smaller and better performing architectures on MNIST-1D than alternative search methods.

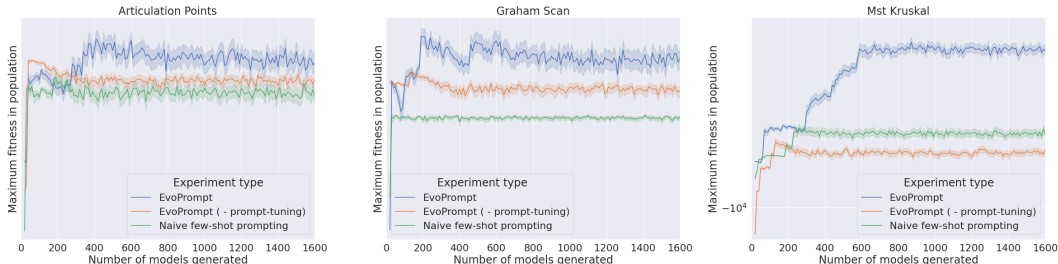

Figure 3: Number of child models generated versus maximum fitness of top model seen so far (as estimated using 100 bootstrap samples of size 20 for each point along the x-axis) when searching over neural network models for three CLRS tasks. As mentioned in Section 4.2, these algorithms were selected because our preliminary analyses indicated that they had the most headroom for architectural improvements.

using as few samples as possible. Figure 2b compares how the fitness of the best-performing child model improves as a function of the number of child samples generated thus far. The random parents baseline plateaus the quickest, reaching a maximum fitness by the time approximately 200 individuals have been generated. Furthermore, the maximum fitness it reaches is significantly worse than that of the other experiments. On the other hand, EVOPROMPTING without prompt-tuning and normal EVOPROMPTING do not plateau until much later on. EVOPROMPTING's plateau is the highest and therefore fitter on average than the individuals discovered by any of the other experiments.

It is also evident from both Figure 2a and 2b that performance suffers when any individual component is removed. Interestingly, Figure 2a indicates that prompting with randomly selected parents combined with prompt-tuning is no more effective than naive prompting alone. This highlights the importance of selecting helpful in-context examples, particularly in a task for which we assume that less training signal exists in the pre-training data. However, selecting more fit models as in-context examples without prompt-tuning also does not perform nearly as well as our full method.

**Trajectory over meta-learning rounds** We also explored the trajectory of our meta-learning algorithm round over round, as shown in Appendix A.3. In general, we observe that EVOPROMPTING starts out further away from the origin (in round 0) and ends up closest to the origin in round 10, which signifies that it discovers – on average – the smallest and most accurate models in the last round. However, the search does not always yield improvements on both axes between consecutive rounds. In rounds 0-2 and 6-10, EVOPROMPTING improves test error while trading off model size. On the other hand, both dimensions are simultaneously improved upon in rounds 3-5.

## 4.2 CLRS

Although the MNIST-1D task offers an efficient and practical setting for evaluating a meta-learning algorithm, CNN architectures already perform fairly well on this task and neural image classification architectures have been extensively studied as a whole. There also exists the possibility that our LM has seen many convolutional architectures in its pre-training data. Instead, we turn to a different learning task and class of neural network architectures in order to assess whether our meta-learning framework generalizes to other tasks, datasets, and neural architectures.

**Dataset**    The CLRS algorithmic reasoning benchmark (Veličković et al., 2022) evaluates the ability of neural networks to learn algorithmic reasoning across a set of 30 classical algorithms covered in the *Introduction to Algorithms* textbook by Cormen, Leiserson, Rivest and Stein (Cormen et al., 2009). This benchmark is useful not only as a difficult logical reasoning task for neural networks, but also as a measure of a neural network's *algorithmic alignment* (Xu et al., 2020). In brief, algorithmic alignment refers to a model's ability to reason like an algorithm (*i.e.* using the computation graph for a task), rather than relying upon memorization or other less sample efficient learning strategies. Although a model can approximate an algorithm by pattern-matching against similar inputs or relying on other shortcuts, it cannot generalize to arbitrarily long inputs or edge cases without learning the computation graph underlying the algorithm.

Accordingly, the CLRS benchmark represents the algorithms' inputs and outputs as graphs, and the steps of the algorithm as a *trajectory* of operations over the input graph. This problem setup can be straightforwardly processed by graph neural networks, which is explored in Ibarz et al. (2022). They find that a Triplet-GMPNN model (a message-passing neural network (Gilmer et al., 2017) with gating and triplet edge processing) exhibits the best performance when trained and evaluated across all 30 algorithms at once.

Table 1: A comparison of OOD accuracy and model size (in number of parameters) of models newly discovered by EVOPROMPTING on select CLRS tasks where EVOPROMPTING has discovered more accurate architectures without large increases in model size, compared with the baseline model (the Triplet-GMPNN from Ibarz et al. (2022)). OOD accuracy numbers for the baseline model are from Ibarz et al. (2022). For the full table of results on all CLRS tasks, including accuracies of our own implementation of the Triplet-GMPNN, see Appendix 3.

| CLRS Task | Best Performing Model | Model Size ↓ | | OOD Accuracy ↑ | |
|---|---|---|---|---|---|
| | | Ours | Baseline | Ours | Baseline |
| Articulation Points | QUADNODEMINMAX | 497969 | 531913 | **93.5 ± 1.8%** | 88.3± 2.0% |
| BFS | MAXMEAN | 522931 | 523963 | **100.0 ± 0.0%** | 99.7± 0.0% |
| Bubble Sort | CONCATREP | 568533 | 524477 | **88.9 ± 2.8%** | 67.7± 5.5% |
| DFS | DIV2MEAN | 660158 | 661190 | **68.1 ± 1.4%** | 47.8± 4.2% |
| Floyd Warshall | CONCATREP | 669145 | 625089 | **61.4 ± 0.8%** | 48.5± 1.0% |
| Heapsort | CONCATREP | 703710 | 659654 | **69.9 ± 4.2%** | 31.0± 5.8% |
| Insertion Sort | DIV2MEAN | 523445 | 524477 | **89.5 ± 2.6%** | 78.1± 4.6% |
| Quicksort | DIV2MEAN | 524727 | 525759 | **85.2 ± 4.3%** | 64.6± 5.1% |
| Task Scheduling | TANHEXPANDTRIPLETS | 262333 | 262333 | **88.2 ± 0.4%** | 87.3± 0.4% |

**Meta-learning set-up**    Similar to our MNIST-1D set-up, we use the AdamW optimizer to train each child model on a single NVIDIA Tesla P100 GPU. However, since most of the explored child models were much larger than the MNIST-1D models, we only trained each child model for 2000 steps. Anecdotally, we observed that the performance of different models often diverged by 2000 steps, which provided sufficient signal for the model search process. We otherwise followed the hyperparameters for single-task training in Ibarz et al. (2022) and evaluated models using validation accuracy.

Unlike our MNIST-1D set-up, we only search over the triplet representations of a Triplet-GMPNN model (see Ibarz et al. (2022) for more details), rather than the entire graph processor. We also seed the search with nine different seed models - each a variant of a Triplet-GMPNN model with a different triplet representation. Each seed triplet representation incorporates a minor tweak of a single component of the original triplet representation designed by Ibarz et al. (2022). These include a fully-connected output layer, a sum aggregation, fully-connected node/edge/graph representations,

a simple linear triplet representation, and a bilinear representation (Mnih & Hinton, 2007). All nine are implemented with Haiku (Hennigan et al., 2020), an object-oriented neural network library for Jax (see Appendix A.5 for the source code of the seed models.)

**Generalizing beyond image classification models** We search using EVOPROMPTING on 3 individual algorithms in the CLRS benchmark – the articulation points, Graham scan, and Kruskal's minimum spanning tree algorithms. We select these algorithms because our preliminary analyses with hand-designed architectures showed that they had the most headroom for improvement, although we found that the discovered architectures transfer well to other CLRS benchmark tasks as well (Appx. 3). Our search results are shown in Figure 3. EVOPROMPTING continues to find models that are more "fit" than our other two baselines, though we observed that the results also show more variation than our results for MNIST-1D did.

**Analyzing newly discovered models** Our search across triplet representations yielded several new designs that we sought to evaluate across all algorithms in the CLRS benchmark. Although these new models were discovered in model searches over single algorithms, they oftentimes generalized to other algorithms that were unseen during the model search. Figure 5 shows the trajectory of validation accuracy during training and Table 1 provides OOD accuracies for these models on a few select algorithms. (We defer the reader to Appendix A.4 for the full source code of each newly discovered model and Table A.6 for the full list of OOD accuracies for every algorithm in the CLRS benchmark.)

We note that the model search suggested several simple but effective changes. For example, instead of taking the maximum of the triplet representation, the QUADNODEMINMAX model uses quadruplet node representations instead of triplets, and it subtracts the minimum of the quad representation from the max instead. CONCATREP represents the node, edge, and graph representations as a concatenation of a projection feedforward layer, and MAXMEAN takes the maximum of the triplet representations prior to taking the mean and passing it through the output dense layer. DIV2MEAN scales each of the node representations by $1/2$ and uses a mean aggregation of the triplet representations instead of the max aggregation. TANHEXPANDTRIPLETS applies additional dimension expansion to the triplet representations and applies a hyperbolic tangent function after the max aggregation. See Appx. A.4 for the full code of each discovered model.

Of the 5 newly discovered models that we chose to analyze, CONCATREP is the only one that increases model size. However, as shown in Table 1, CONCATREP frequently yielded improvements in OOD accuracy that far exceeded the percent increase in model size. For instance, on the heapsort algorithm CONCATREP increased OOD accuracy by 125.19% while only increasing model size by 6.68% over the baseline. The other four newly discovered models shown in Table 1 simultaneously improved OOD accuracy while decreasing model size on the articulation points, BFS, DFS, insertion sort, quicksort, and task scheduling algorithms. On the rest of the CLRS algorithms (Table A.6), our newly discovered models typically achieved OOD accuracy comparable to or better than the baseline, while maintaining similar model size.

## 5 Conclusion

We have shown that embedding a pre-trained LM in an evolutionary algorithm significantly improves the LM's performance on the task of neural architecture design. Our approach has demonstrated success at not only optimizing convolutional architectures for the MNIST-1D task, but also at developing new kinds of GNNs for the CLRS algorithmic benchmark. This demonstrates: 1) using evolutionary techniques can vastly improve the in-context capabilities of pre-trained LMs, and 2) EVOPROMPTING can discover novel and state-of-the-art architectures that optimize for both accuracy and model size. Furthermore, EVOPROMPTING is general enough to be easily adapted to search for solutions to other kinds of reasoning tasks beyond NAS. We leave the adaptation of EVOPROMPTING for other tasks to future work.

However, our study is limited by the lack of an extensive comparison against other standard NAS techniques because EVOPROMPTING was designed for open-ended search, whereas other techniques were not, which would introduce a potential confounder. We include one such comparison on NATS-Bench in Appendix A.7, as well as a discussion of the confounders thereof.

# 6 Acknowledgements

We thank Maarten Bosma, Kefan Xiao, Yifeng Lu, Quoc Le, Ed Chi, Borja Ibarz, Petar Veličković, Chen Liang, Charles Sutton, and the Google Brain AutoML team for providing valuable discussions and feedback that influenced the direction of this project. We also thank the Google Student Researcher program for providing the resources and opportunities necessary for this project to take place.

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

# A Appendix

## A.1 EVOPROMPTING Hyperparameters

Table 2: Values of hyperparameters used in EVOPROMPT.

| HYPERPARAMETER | DESCRIPTION | VALUE |
|:---:|:---|:---:|
| $p$ | Num. parents to select in every generation | 10 |
| $k$ | Num. in-context examples in prompt | 2 |
| $T$ | Num. rounds of evolution | 10 |
| $m$ | Num. prompts per round | 10 |
| $n$ | Num. samples to generate per prompt | 16 |
| $\alpha$ | Lower threshold for test error | 0.5 |

## A.2 MNIST-1D Seed Models

Below we provide the source code for the four seed models used in the MNIST-1D model search.

```python
class Model(nn.Module):
  features: int = 32
  nlayer: int = 3

  @nn.compact
  def __call__(self, x):
    x = x[..., None]
    x = nn.Conv(features=self.features, kernel_size=(3,))(x)
    x = nn.relu(x)

    x = nn.avg_pool(x, window_shape=(2,), strides=(2,))
    for _ in range(self.nlayer - 1):
      xp = nn.Conv(
          features=self.features,
          kernel_size=(3,),
      )(x)
      xp = nn.relu(xp)
      x = x + xp

    x = nn.avg_pool(x, window_shape=(2,), strides=(2,))
    x = x.reshape((x.shape[0], -1))   # flatten
    x = nn.Dense(features=256)(x)
    x = nn.relu(x)
    x = nn.Dense(features=10)(x)
    return x
```

Listing 2: A hand-designed convolutional model.

```python
class Model(nn.Module):
  features: int = 25

  @nn.compact
  def __call__(self, x):
    x = x[..., None]
    x = nn.Conv(
        features=self.features, kernel_size=(5,), strides=(2,),
    padding=(1,)
    )(x)
    x = nn.relu(x)
    for _ in range(2):
```

```
12        x = nn.Conv(
13            features=self.features, kernel_size=(3,), strides=(2,)
    , padding=(1,)
14        )(x)
15        x = nn.relu(x)
16      x = x.reshape((x.shape[0], -1))
17      x = nn.Dense(features=10)(x)
18      return x
```

Listing 3: A Flax implementation of the convolutional baseline from Greydanus (2020).

```
1 class Model(nn.Module):
2   """A simple GRU model."""
3
4   hidden_size: int = 6
5   seed: int = 42
6
7   @nn.compact
8   def __call__(self, x):
9     x = jnp.expand_dims(x, -1)
10     rng = jax_random.PRNGKey(self.seed)
11     gru = recurrent.GRU(
12         hidden_size=self.hidden_size,
13         num_layers=1,
14         dropout_rate=0.0,
15         bidirectional=True,
16     )
17     lengths = np.full([x.shape[0]], x.shape[1])
18     initialized_params = gru.init(rng, x, lengths)
19     params = initialized_params['params']
20     outputs, _ = gru.apply({'params': params}, x, lengths)
21     outputs = outputs.reshape((outputs.shape[0], -1))
22     x = nn.Dense(features=10)(outputs)
23     return x
```

Listing 4: A Flax implementation of the GRU baseline from Greydanus (2020).

```
1 class Model(nn.Module):
2   hidden_size: int = 100
3
4   @nn.compact
5   def __call__(self, x):
6     x = nn.Dense(features=self.hidden_size)(x)
7     x = nn.relu(x)
8     x = x + nn.relu(nn.Dense(features=self.hidden_size)(x))
9     x = nn.Dense(features=10)(x)
10     return x
11
12 return Model
```

Listing 5: A Flax implementation of the fully connected baseline from Greydanus (2020).

## A.3  Trajectory of search for MNIST-1D models

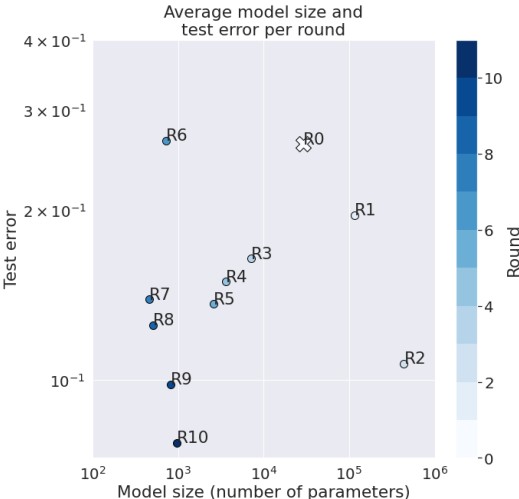

Figure 4: The average model size and test error of the child models produced in each round of the model search. Data points closer to the origin represent rounds that yielded more "fit" models.

## A.4 Newly Discovered CLRS GNNs

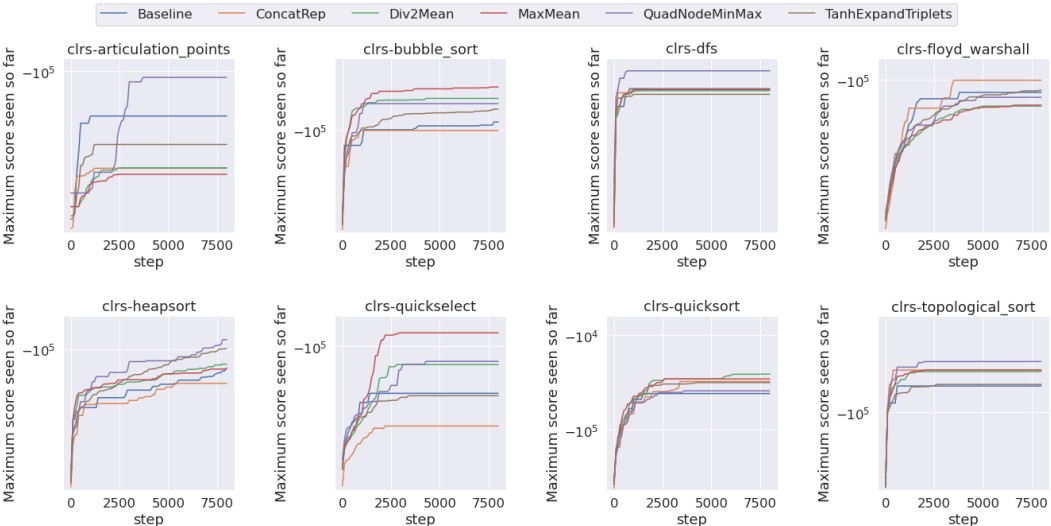

Figure 5: Maximum fitness scores of five of the newly discovered models, compared against the baseline, on eight of the CLRS tasks.

Below we list the Python source code of five of the newly discovered GNNs.

```python
def get_triplet_msgs_quad(z, edge_fts, graph_fts, nb_triplet_fts
    , out_size):
  node_reps = [hk.Linear(nb_triplet_fts) for _ in range(4)]
  triplet_node_reps = [node_rep(z) for node_rep in node_reps]
  node_pair_inversions = [(1, 2), (1, 3), (2, 3), (3, 1)]
  triplets = functools.reduce(
      lambda x, y: x + y,
      [
          jnp.expand_dims(tri_node_rep, axis=perm)
          for tri_node_rep, perm in zip(
```

```
10                    triplet_node_reps, node_pair_inversions
11               )
12          ],
13    )
14    return jnp.max(triplets, axis=1) - jnp.min(triplets, axis=1)
```

Listing 6: The triplet representation that we refer to as QUADNODEMINMAX.

```
1 def get_triplet_msgs_concatrep(z, edge_fts, graph_fts,
      nb_triplet_fts, out_size):
2   def rep_fn(x, size):
3     proj = hk.nets.MLP([size])
4     ff = hk.nets.MLP([size * 4, size])
5     return jnp.concatenate([
6       proj(x),
7       ff(x),
8     ], axis=-1)
9
10   triplet_node_reps = [rep_fn(z, nb_triplet_fts) for _ in range
      (3)]
11   triplet_edge_reps = [rep_fn(edge_fts, nb_triplet_fts) for _ in
       range(3)]
12   triplet_graph_rep = rep_fn(graph_fts, nb_triplet_fts)
13   node_pair_permutations = [(2, 3), (1, 3), (1, 2)]
14   triplets = functools.reduce(
15       lambda x, y: x + y,
16       [
17           jnp.expand_dims(tri_node_rep, axis=perm)
18           for tri_node_rep, perm in zip(
19               triplet_node_reps, node_pair_permutations
20           )
21       ],
22   )
23   triplets += functools.reduce(
24       lambda x, y: x + y,
25       [
26           jnp.expand_dims(tri_edge_rep, axis=i)
27           for tri_edge_rep, i in zip(triplet_edge_reps, range(3,
      0, -1))
28       ],
29   )
30   triplets += jnp.expand_dims(triplet_graph_rep, axis=(1, 2, 3))
31   output_layer = hk.Linear(out_size)
32   return output_layer(jnp.max(triplets, axis=1))
```

Listing 7: The triplet representation that we refer to as CONCATREP.

```
1 def get_triplet_msgs_tanhexplandtriplets(z, edge_fts, graph_fts,
       nb_triplet_fts, out_size):
2   node_reps = [hk.Linear(nb_triplet_fts) for _ in range(3)]
3   edge_reps = [hk.Linear(nb_triplet_fts) for _ in range(3)]
4   graph_rep = hk.nets.MLP([nb_triplet_fts])
5   triplet_node_reps = [node_rep(z) for node_rep in node_reps]
6   triplet_edge_reps = [edge_rep(edge_fts) for edge_rep in
      edge_reps]
7   node_pair_permutations = [(2, 3), (1, 3), (1, 2)]
8   triplets = functools.reduce(
9       lambda x, y: x + y,
10       [
11           jnp.expand_dims(tri_node_rep, axis=perm)
12           for tri_node_rep, perm in zip(
```

```
13                  triplet_node_reps , node_pair_permutations
14              )
15          ],
16      )
17      triplets += functools.reduce(
18          lambda x, y: x + y,
19          [
20              jnp.expand_dims(tri_edge_rep, axis=i)
21              for tri_edge_rep, i in zip(triplet_edge_reps, range(3,
        0, -1))
22          ],
23      )
24      triplets += jnp.expand_dims(graph_rep(graph_fts), axis=(1, 2,
        3))
25      triplets += jnp.expand_dims(graph_rep(graph_fts), axis=(2, 3,
        1))
26      triplets += jnp.expand_dims(graph_rep(graph_fts), axis=(3, 1,
        2))
27      output_layer = hk.Linear(out_size)
28      return output_layer(jnp.tanh(jnp.max(triplets, axis=1)))
```

Listing 8: The triplet representation that we refer to as TANHEXPANDTRIPLETS.

```
1 def get_triplet_msgs_div2mean(z, edge_fts, graph_fts,
      nb_triplet_fts, out_size):
2   node_reps = [hk.Linear(nb_triplet_fts) for _ in range(3)]
3   edge_reps = [hk.Linear(nb_triplet_fts) for _ in range(3)]
4   triplet_node_reps = [node_rep(z / 2) for node_rep in node_reps
      ]
5   triplet_edge_reps = [edge_rep(edge_fts) for edge_rep in
      edge_reps]
6   node_pair_permutations = [(2, 3), (1, 3), (1, 2)]
7   triplets = functools.reduce(
8       lambda x, y: x + y,
9       [
10           jnp.expand_dims(tri_node_rep, axis=perm )
11           for tri_node_rep, perm in zip(
12               triplet_node_reps, node_pair_permutations
13           )
14       ],
15   )
16   triplets += functools.reduce(
17       lambda x, y: x + y,
18       [
19           jnp.expand_dims(tri_edge_rep, axis=perm)
20           for tri_edge_rep, perm in zip(triplet_edge_reps, range
      (3, 0, -1))
21       ],
22   )
23   output_layer = hk.Linear(out_size)
24   return output_layer(jnp.mean(triplets, axis=1))
```

Listing 9: The triplet representation that we refer to as DIV2MEAN.

```
1 def get_triplet_msgs_maxmean(z, edge_fts, graph_fts,
      nb_triplet_fts, out_size):
2   node_reps = [hk.Linear(nb_triplet_fts) for _ in range(3)]
3   edge_reps = [hk.Linear(nb_triplet_fts) for _ in range(3)]
4   graph_rep = hk.nets.MLP([nb_triplet_fts])
5   triplet_node_reps = [node_rep(z) for node_rep in node_reps]
```

```
6    triplet_edge_reps = [edge_rep(edge_fts) for edge_rep in
         edge_reps]
7    node_pair_permutations = [(2, 3), (1, 3), (1, 2)]
8    triplets = functools.reduce(
9        lambda x, y: x + y,
10       [
11           jnp.expand_dims(tri_node_rep, axis=perm)
12           for tri_node_rep, perm in zip(
13               triplet_node_reps, node_pair_permutations
14           )
15       ],
16   )
17   triplets += functools.reduce(
18       lambda x, y: x + y,
19       [
20           jnp.expand_dims(tri_edge_rep, axis=i)
21           for tri_edge_rep, i in zip(triplet_edge_reps, range(3,
         0, -1))
22       ],
23   )
24   triplets = jnp.maximum(triplets, -100.0)
25   output_layer = hk.Linear(out_size)
26   return output_layer(jnp.mean(triplets, axis=1))
```

Listing 10: The triplet representation that we refer to as MAXMEAN.

## A.5 CLRS Seed Models

Below we provide the source code for the nine seed models used in the CLRS model search.

```python
def get_triplet_msgs_v1(z, edge_fts, graph_fts, nb_triplet_fts,
    out_size):
  node_reps = [hk.Linear(nb_triplet_fts) for _ in range(3)]
  edge_reps = [hk.Linear(nb_triplet_fts) for _ in range(3)]
  graph_rep = hk.Linear(nb_triplet_fts)
  triplet_node_reps = [node_rep(z) for node_rep in node_reps]
  triplet_edge_reps = [edge_rep(edge_fts) for edge_rep in
    edge_reps]
  triplet_graph_rep = graph_rep(graph_fts)
  node_pair_permutations = [(2, 3), (1, 3), (1, 2)]
  triplets = functools.reduce(
      lambda x, y: x + y,
      [
          jnp.expand_dims(tri_node_rep, axis=perm)
          for tri_node_rep, perm in zip(
              triplet_node_reps, node_pair_permutations
          )
      ],
  )
  triplets += functools.reduce(
      lambda x, y: x + y,
      [
          jnp.expand_dims(tri_edge_rep, axis=i)
          for tri_edge_rep, i in zip(triplet_edge_reps, range(3,
    0, -1))
        ],
  )
  triplets += jnp.expand_dims(triplet_graph_rep, axis=(1, 2, 3))
  output_layer = hk.Linear(out_size)
  return output_layer(jnp.max(triplets, axis=1))
```

Listing 11: The triplet representation belonging to the first seed model - the standard triplet representation from Ibarz et al. (2022).

```python
def get_triplet_msgs_v2(z, edge_fts, graph_fts, nb_triplet_fts,
    out_size):
  node_reps = [hk.Linear(nb_triplet_fts) for _ in range(3)]
  edge_reps = [hk.Linear(nb_triplet_fts) for _ in range(3)]
  graph_rep = hk.Linear(nb_triplet_fts)
  triplet_node_reps = [node_rep(z) for node_rep in node_reps]
  triplet_edge_reps = [edge_rep(edge_fts) for edge_rep in
    edge_reps]
  triplet_graph_rep = graph_rep(graph_fts)
  node_pair_permutations = [(2, 3), (1, 3), (1, 2)]
  triplets = functools.reduce(
      lambda x, y: x + y,
      [
          jnp.expand_dims(tri_node_rep, axis=perm)
          for tri_node_rep, perm in zip(
              triplet_node_reps, node_pair_permutations
          )
      ],
  )
  triplets += functools.reduce(
      lambda x, y: x + y,
      [
          jnp.expand_dims(tri_edge_rep, axis=i)
          for tri_edge_rep, i in zip(triplet_edge_reps, range(3,
    0, -1))
```

```
23        ],
24    )
25    triplets += jnp.expand_dims(triplet_graph_rep, axis=(1, 2, 3))
26    output_layer = hk.nets.MLP([out_size, out_size])
27    return output_layer(jnp.max(triplets, axis=1))
```

Listing 12: The triplet representation belonging to the second seed model, with the output layer replaced by a fully-connected multi-layer perceptron.

```
1  def get_triplet_msgs_v3(z, edge_fts, graph_fts, nb_triplet_fts,
       out_size):
2    node_reps = [hk.Linear(nb_triplet_fts) for _ in range(3)]
3    edge_reps = [hk.Linear(nb_triplet_fts) for _ in range(3)]
4    graph_rep = hk.Linear(nb_triplet_fts)
5    triplet_node_reps = [node_rep(z) for node_rep in node_reps]
6    triplet_edge_reps = [edge_rep(edge_fts) for edge_rep in
       edge_reps]
7    triplet_graph_rep = graph_rep(graph_fts)
8    node_pair_permutations = [(2, 3), (1, 3), (1, 2)]
9    triplets = functools.reduce(
10       lambda x, y: x + y,
11       [
12           jnp.expand_dims(tri_node_rep, axis=perm)
13           for tri_node_rep, perm in zip(
14               triplet_node_reps, node_pair_permutations
15           )
16       ],
17   )
18   triplets += functools.reduce(
19       lambda x, y: x + y,
20       [
21           jnp.expand_dims(tri_edge_rep, axis=i)
22           for tri_edge_rep, i in zip(triplet_edge_reps, range(3,
       0, -1))
23       ],
24   )
25   triplets += jnp.expand_dims(triplet_graph_rep, axis=(1, 2, 3))
26   output_layer = hk.Linear(out_size)
27   return output_layer(jnp.sum(triplets, axis=1))
```

Listing 13: The triplet representation belonging to the third seed model, which uses sum instead of max aggregation.

```
1  def get_triplet_msgs_v4(z, edge_fts, graph_fts, nb_triplet_fts,
       out_size):
2    node_reps = [hk.nets.MLP([nb_triplet_fts, nb_triplet_fts]) for
       _ in range(3)]
3    edge_reps = [hk.Linear(nb_triplet_fts) for _ in range(3)]
4    graph_rep = hk.Linear(nb_triplet_fts)
5    triplet_node_reps = [node_rep(z) for node_rep in node_reps]
6    triplet_edge_reps = [edge_rep(edge_fts) for edge_rep in
       edge_reps]
7    triplet_graph_rep = graph_rep(graph_fts)
8    node_pair_permutations = [(2, 3), (1, 3), (1, 2)]
9    triplets = functools.reduce(
10       lambda x, y: x + y,
11       [
12           jnp.expand_dims(tri_node_rep, axis=perm)
13           for tri_node_rep, perm in zip(
14               triplet_node_reps, node_pair_permutations
15           )
```

```
16          ],
17      )
18      triplets += functools.reduce(
19          lambda x, y: x + y,
20          [
21              jnp.expand_dims(tri_edge_rep, axis=i)
22              for tri_edge_rep, i in zip(triplet_edge_reps, range(3,
      0, -1))
23          ],
24      )
25      triplets += jnp.expand_dims(triplet_graph_rep, axis=(1, 2, 3))
26      output_layer = hk.Linear(out_size)
27      return output_layer(jnp.max(triplets, axis=1))
```

Listing 14: The triplet representation of the 4th seed model, which uses fully-connected multi-layer perceptron node representations.

```
1  def get_triplet_msgs_v5(z, edge_fts, graph_fts, nb_triplet_fts,
       out_size):
2      node_reps = [hk.Linear(nb_triplet_fts) for _ in range(3)]
3      edge_reps = [hk.nets.MLP([nb_triplet_fts, nb_triplet_fts]) for
       _ in range(3)]
4      graph_rep = hk.Linear(nb_triplet_fts)
5      triplet_node_reps = [node_rep(z) for node_rep in node_reps]
6      triplet_edge_reps = [edge_rep(edge_fts) for edge_rep in
       edge_reps]
7      triplet_graph_rep = graph_rep(graph_fts)
8      node_pair_permutations = [(2, 3), (1, 3), (1, 2)]
9      triplets = functools.reduce(
10         lambda x, y: x + y,
11         [
12             jnp.expand_dims(tri_node_rep, axis=perm)
13             for tri_node_rep, perm in zip(
14                 triplet_node_reps, node_pair_permutations
15             )
16         ],
17     )
18     triplets += functools.reduce(
19         lambda x, y: x + y,
20         [
21             jnp.expand_dims(tri_edge_rep, axis=i)
22             for tri_edge_rep, i in zip(triplet_edge_reps, range(3,
      0, -1))
23         ],
24     )
25     triplets += jnp.expand_dims(triplet_graph_rep, axis=(1, 2, 3))
26     output_layer = hk.Linear(out_size)
27     return output_layer(jnp.max(triplets, axis=1))
```

Listing 15: The triplet representation of the 5th seed model, which uses fully-connected multi-layer perceptron edge representations.

```
1  def get_triplet_msgs_v6(z, edge_fts, graph_fts, nb_triplet_fts,
       out_size):
2      node_reps = [hk.Linear(nb_triplet_fts) for _ in range(3)]
3      edge_reps = [hk.Linear(nb_triplet_fts) for _ in range(3)]
4      graph_rep = hk.nets.MLP([nb_triplet_fts, nb_triplet_fts])
5      triplet_node_reps = [node_rep(z) for node_rep in node_reps]
6      triplet_edge_reps = [edge_rep(edge_fts) for edge_rep in
       edge_reps]
7      triplet_graph_rep = graph_rep(graph_fts)
```

```
8    node_pair_permutations = [(2, 3), (1, 3), (1, 2)]
9    triplets = functools.reduce(
10       lambda x, y: x + y,
11       [
12           jnp.expand_dims(tri_node_rep, axis=perm)
13           for tri_node_rep, perm in zip(
14               triplet_node_reps, node_pair_permutations
15           )
16       ],
17   )
18   triplets += functools.reduce(
19       lambda x, y: x + y,
20       [
21           jnp.expand_dims(tri_edge_rep, axis=i)
22           for tri_edge_rep, i in zip(triplet_edge_reps, range(3,
    0, -1))
23         ],
24   )
25   triplets += jnp.expand_dims(triplet_graph_rep, axis=(1, 2, 3))
26   output_layer = hk.Linear(out_size)
27   return output_layer(jnp.max(triplets, axis=1))
```

Listing 16: The triplet representation of the 6th seed model, which uses fully-connected multi-layer perceptron graph representations.

```
1 def get_triplet_msgs_v7(z, edge_fts, graph_fts, nb_triplet_fts,
      out_size):
2   node_reps = [hk.nets.MLP([nb_triplet_fts]) for _ in range(3)]
3   edge_reps = [hk.Linear(nb_triplet_fts) for _ in range(3)]
4   triplet_node_reps = [node_rep(z) for node_rep in node_reps]
5   triplet_edge_reps = [edge_rep(edge_fts) for edge_rep in
      edge_reps]
6   node_pair_permutations = [(2, 3), (1, 3), (1, 2)]
7   triplets = functools.reduce(
8       lambda x, y: x + y,
9       [
10          jnp.expand_dims(tri_node_rep, axis=perm)
11          for tri_node_rep, perm in zip(
12              triplet_node_reps, node_pair_permutations
13          )
14      ],
15  )
16  triplets += functools.reduce(
17      lambda x, y: x + y,
18      [
19          jnp.expand_dims(tri_edge_rep, axis=i)
20          for tri_edge_rep, i in zip(triplet_edge_reps, range(3,
    0, -1))
21        ],
22  )
23  output_layer = hk.Linear(out_size)
24  return output_layer(jnp.max(triplets, axis=1))
```

Listing 17: The triplet representation of the 7th seed model, which uses fully-connected multi-layer perceptron node representations and does not have a graph representation.

```
1 def get_triplet_msgs_v8(z, edge_fts, graph_fts, nb_triplet_fts,
      out_size):
2   output_layer = hk.nets.MLP([out_size])
3   return jnp.tile(
```

```
4          jnp.expand_dims(output_layer(z), axis=(1)), [1, z.shape
      [1], 1, 1]
5    )
```

Listing 18: The triplet representation of the 8th seed model, which simply applies a linear layer and tiles the output to maintain dimensional consistency.

```
1 def get_triplet_msgs_v9(z, edge_fts, graph_fts, nb_triplet_fts,
      out_size):
2    def rep_fn(x, size):
3      proj = hk.nets.MLP([size])
4      ff = hk.nets.MLP([size * 8, size])
5      return proj(x) * ff(x)
6
7    triplet_node_reps = [rep_fn(z, nb_triplet_fts) for _ in range
      (3)]
8    triplet_edge_reps = [rep_fn(edge_fts, nb_triplet_fts) for _ in
       range(3)]
9    triplet_graph_rep = rep_fn(graph_fts, nb_triplet_fts)
10   node_pair_permutations = [(2, 3), (1, 3), (1, 2)]
11   triplets = functools.reduce(
12       lambda x, y: x + y,
13       [
14           jnp.expand_dims(tri_node_rep, axis=perm)
15           for tri_node_rep, perm in zip(
16               triplet_node_reps, node_pair_permutations
17           )
18       ],
19   )
20   triplets += functools.reduce(
21       lambda x, y: x + y,
22       [
23           jnp.expand_dims(tri_edge_rep, axis=i)
24           for tri_edge_rep, i in zip(triplet_edge_reps, range(3,
      0, -1))
25       ],
26   )
27   triplets += jnp.expand_dims(triplet_graph_rep, axis=(1, 2, 3))
28   return rep_fn(jnp.max(triplets, axis=1), out_size)
```

Listing 19: The triplet representation of the 9th seed model, which uses a bilinear representation for the node, edge, and graph representations.

## A.6  OOD Evaluation of Newly Discovered Models on CLRS

Table 3: A full list comparing 5 of the newly discovered GNNs against the baseline Triplet-GMPNN model from Ibarz et al. (2022) on all 30 of the CLRS (Veličković et al., 2022) tasks. For the baseline, we include the OOD accuracy of both our implementation of the Triplet-GMPNN as well as the number listed in the original paper.

| Algorithm | Best Performing Model | Model Size ↓ | | OOD Accuracy ↑ | | |
| --- | --- | --- | --- | --- | --- | --- |
| | | Best Performing Model | Baseline model | Best performing newly discovered model | Baseline model (our implementation) | Baseline model (from Ibarz et al. (2022)) |
| Activity Selector | Baseline | 262204 | 262204 | 95.05 ± 0.53% | 93.96± 0.29% | 95.18± 0.45% |
| Articulation Points | QUADNODEMINMAX | 497969 | 531913 | 93.46 ± 1.77% | 91.40± 1.74% | 88.32± 2.01% |
| Bellman Ford | CONCATREP | 568660 | 524604 | 97.50 ± 0.31% | 97.08± 0.24% | 97.39± 0.19% |
| BFS | MAXMEAN | 522931 | 523963 | 99.99 ± 0.01% | 99.80± 0.04% | 99.73± 0.04% |
| Binary Search | Baseline | 262204 | 262204 | 77.98 ± 2.49% | 79.57± 1.73% | 77.58± 2.35% |
| Bridges | CONCATREP | 576612 | 532556 | 97.57 ± 1.08% | 97.31± 1.11% | 93.99± 2.07% |
| Bubble Sort | CONCATREP | 568533 | 524477 | 88.87 ± 2.77% | 83.20± 4.27% | 67.68± 5.50% |
| DAG Shortest Paths | Baseline | 793287 | 793287 | 98.01 ± 0.22% | 97.48± 0.37% | 98.19± 0.30% |
| DFS | DIV2MEAN | 660158 | 661190 | 68.14 ± 1.38% | 46.78± 3.85% | 47.79± 4.19% |
| Dijkstra | DIV2MEAN | 524854 | 525886 | 97.30 ± 0.28% | 95.94± 0.66% | 96.05± 0.60% |
| Find Maximum Subarray Kadane | Baseline | 261290 | 264514 | 75.35 ± 0.92% | 74.09± 0.83% | 76.36± 0.43% |
| Floyd Warshall | CONCATREP | 669145 | 625089 | 61.43 ± 0.79% | 48.95± 0.49% | 48.52± 1.04% |
| Graham Scan | MAXMEAN | 397377 | 398409 | 93.76 ± 0.85% | 92.72± 2.38% | 93.62± 0.91% |
| Heapsort | CONCATREP | 703710 | 659654 | 69.90 ± 4.17% | 19.45± 5.35% | 31.04± 5.82% |
| Insertion Sort | DIV2MEAN | 523445 | 524477 | 89.47 ± 2.57% | 86.89± 1.89% | 78.14± 4.64% |
| Jarvis March | Baseline | 308954 | 264898 | 90.36 ± 0.65% | 88.91± 0.91% | 91.01± 1.30% |
| Knuth-Morris-Pratt | Baseline | 396989 | 398021 | 16.29 ± 4.36% | 8.88± 1.76% | 19.51± 4.57% |
| LCS Length | Baseline | 270419 | 270419 | 85.75 ± 0.80% | 86.05± 0.65% | 80.51± 1.84% |
| Matrix Chain Order | Baseline | 624448 | 624448 | 90.77 ± 0.75% | 91.15± 0.85% | 91.68± 0.59% |
| Minimum | DIV2MEAN | 260275 | 261307 | 98.40 ± 0.16% | 98.26± 0.26% | 97.78± 0.55% |

| Algorithm | Best Performing Model | Model Size ↓ | | OOD Accuracy ↑ | | |
| --- | --- | --- | --- | --- | --- | --- |
| | | Best Performing Model | Baseline model | Best performing newly discovered model | Baseline model (our implementation) | Baseline model (from Ibarz et al. (2022)) |
| MST Kruskal | CONCATREP | 443747 | 399691 | 91.47 ± 0.48% | 90.60± 0.32% | 89.80± 0.77% |
| MST Prim | CONCATREP | 569942 | 525886 | 88.74 ± 1.67% | 85.18± 2.24% | 86.39± 1.33% |
| Naive String Matcher | QUADNODEMINMAX | 259364 | 262588 | 79.77 ± 2.88% | 73.39± 6.33% | 78.67± 4.99% |
| Optimal BST | DIV2MEAN | 624955 | 625987 | 78.66 ± 0.46% | 78.08± 0.96% | 73.77± 1.48% |
| Quickselect | QUADNODEMINMAX | 377130 | 395714 | 0.79 ± 0.41% | 0.13± 0.08% | 0.47± 0.25% |
| Quicksort | DIV2MEAN | 524727 | 525759 | 85.23 ± 4.26% | 84.71± 2.66% | 64.64± 5.12% |
| Segments Intersect | DIV2MEAN | 262327 | 263359 | 98.15 ± 0.00% | 97.40± 0.00% | 97.64± 0.09% |
| Strongly Connected Components | Baseline | 707299 | 663243 | 41.86 ± 3.39% | 43.71± 5.94% | 43.43± 3.15% |
| Task Scheduling | TANHEXPANDTRIPLETS | 262333 | 262333 | 88.23 ± 0.44% | 88.10± 0.31% | 87.25± 0.35% |
| Topological Sort | TANHEXPANDTRIPLETS | 660164 | 660164 | 88.12 ± 4.71% | 76.88± 5.05% | 87.27± 2.67% |

End of Table 3

## A.7 NATS-Bench

We make a brief comparison of EVOPROMPTING against other NAS algorithms on the NATS-Bench Size Search Space (Dong et al., 2021). However, we note that this comparison is limited because it handicaps EVOPROMPTING in multiple ways:

1. Our LLM was pre-trained to generate code, but for this comparison it is prompted to generate the NATS-Bench style architecture strings, which are in the format "64:64:64:64:64," which removes the benefit of its code pre-training.

2. One of EVOPROMPTING main advantages is its open-ended search space. Since its search space does not need to be hand-designed, EVOPROMPTING can potentially discover novel architectures that other NAS algorithms cannot.

Table 4: EVOPROMPTING versus other standard neural architecture search algorithms on the NATS-Bench size search space. Since properly tuning the hyperparameter values for all search techniques is non-trivial, we obtain the accuracy numbers for all methods other than ours from Dong et al. (2021). EVOPROMPTING performs competitively compared to all the other techniques.

|  | Method | CIFAR-10 | | CIFAR-100 | | ImageNet-16-120 | |
|  |  | Val | Test | Val | Test | Val | Test |
|---|---|---|---|---|---|---|---|
| Multi-trial | REA | 90.37± 0.20 | 93.22± 0.16 | 70.23± 0.50 | 70.11± 0.61 | 45.30± 0.69 | 45.94± 0.92 |
|  | REINFORCE | 90.25± 0.23 | 93.16± 0.21 | 69.84± 0.59 | 69.96± 0.57 | 45.06± 0.77 | 45.71± 0.93 |
|  | RANDOM | 90.10± 0.26 | 93.03± 0.25 | 69.57± 0.57 | 69.72± 0.61 | 45.01± 0.74 | 45.42± 0.86 |
|  | BOHB | 90.07± 0.28 | 93.01± 0.24 | 69.75± 0.60 | 69.90± 0.60 | 45.11± 0.69 | 45.56± 0.81 |
| Weight-sharing | channel-wise interpolation | 90.71± 0.00 | 93.40± 0.00 | 70.30± 0.00 | 70.72± 0.00 | 44.73± 0.00 | 47.17± 0.00 |
|  | masking + Gumbel-Softmax | 90.41± 0.10 | 93.14± 0.13 | 70.30± 0.00 | 70.72± 0.00 | 45.71± 0.39 | 46.38± 0.27 |
|  | masking + sampling | 89.73± 0.37 | 92.78± 0.30 | 69.67± 0.22 | 70.11± 0.33 | 44.70± 0.60 | 45.11± 0.76 |
|  | EVOPROMPT-ING | 90.38± 0.33 | 93.11± 0.90 | 70.47± 0.23 | 70.39± 0.58 | 45.32± 0.26 | 45.15± 0.51 |

## A.8 Broader Impacts

Our work may have a number of ethical, societal, and other broader impacts. Since we focus on automatic improvement of large language models, the implications of our research are largely similar to those of LMs in general. On the one hand, improving the abilities and decreasing the sizes of LMs may increase their accessibility (Köpf et al., 2023), improve energy efficiency (McDonald et al., 2022; Chen et al., 2023), and expand educational and professional opportunities (Kasneci et al., 2023; Eysenbach, 2023). On the other, LMs have long been known to give rise to unjust and toxic language that may hurt and amplify stereotypes (Nadeem et al., 2020; Lucy & Bamman, 2021), exclusionary norms, and allocational harms to marginalized groups (Bender et al., 2021). LMs may also present information hazards, often generating realistic-sounding misinformation (Bickmore et al., 2018; Quach, 2022) or revealing private personal information (Carlini et al., 2021). Lastly, other harms may arise from the ways that humans interact with LMs – either by inadvertently relying too much on unreliable LM outputs (McKee et al., 2021) or via malicious uses (Ranade et al., 2021; Boiko et al., 2023).

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
