# OpenReview forum: "EvoPrompting: Language Models for Code-Level Neural Architecture Search"
_NeurIPS.cc/2023/Conference — NeurIPS 2023 poster_

### Official Review · Reviewer_ziXe · 2023-06-18

**Soundness:** 1 poor
**Presentation:** 2 fair
**Contribution:** 1 poor
**Rating:** 4
**Confidence:** 5

**Summary:**

This paper proposed an interesting idea for neural architecture search. Specifically, the idea is based on evolutionary computation and an LLM, where the LLM plays the role of genetic search, i.e., crossover operation and mutation operation. The experiments are conducted on two benchmarks: MNIST-1D and CLRS, which are for convolutional network networks and graph neural networks, respectively.
===================================
Thanks very much for the authors' effort in explaining. As can be seen, the necessary ablation study is still not provided for the justification. However, considering the idea is interesting in this work, I will adjust my score to be 'Borderline reject'.

**Strengths:**

The LLM is explored for the neural architecture search.

**Weaknesses:**

The experiments can only serve the purpose of demonstration, while the usefulness of the idea is not clear. The main problem is the experiments are not common ones within the community of neural architecture search, and the compared algorithms are not the state of the arts in the community. The use of LLM plays the role of crossover and mutation, while the ablation studies are missing for the verification of LLM better than the original crossover operator and mutation operator. In addition, the fitness function is quite odd because it is obtained through the test from the authors, while the motivation is not clear.


**Questions:**

This paper mentioned multiple times that existing evolutionary approaches for neural architecture search have required careful design and specification of a discrete search space, while the proposed method includes any neural network architecture. I am not sure of the particular meaning of: "any" in this claim. However, I would like to say that the references mentioned for the evolution approaches just used fixed-length encoding, so they may have that limitation. In addition, there is also some work [1] using evolutionary approaches for neural architecture search with variable-length encoding, the search space also includes any neural network architectures.

[1] Sun et. al, “Evolving deep convolutional neural networks for image classification,” IEEE Transactions on Evolutionary Computation, vol. 24, no. 2, pp. 394-407, 2020.

**Limitations:**

See Weaknesses.

---

> ### Author Rebuttal · Authors · 2023-08-10
>
> We thank the reviewer for taking the time and energy to read our paper and provide feedback. Below, we provide responses to the main concerns and questions raised in your review.
>
> "I am not sure of the particular meaning of: "any" in this claim."
> - Sorry for the confusion – we use "any" to distinguish EvoPrompting from existing NAS methods, which are constrained by fixed operator vocabularies and patterns for combining those operators. For example, a common NAS practice is to have an "activation" operator (e.g. swish, GELU, ReLU) that can take N possible values. In contrast, EvoPrompting does not have this restriction and can output python code that encodes other activation functions as well, even ones that may not have been used before in other architectures.
> - In reference to the issue of fixed- vs. variable-length encoding, our approach is indeed limited by the context length of the LM, and we are happy to add this caveat to the paper. (Thank you!) However, this limitation may be mitigated in a couple of ways: 1) the context lengths of modern LMs are constantly increasing as progress in the field continues rapidly, and 2) architectures are modular - EvoPrompting may be applied sequentially or in parallel to optimize individual sub-components of architectures, just as we did in Section 4.2 for the triplet processors of GNNs.
>
> "The experiments can only serve the purpose of demonstration, while, the usefulness of the idea is not clear"
> - EvoPrompting provides multiple benefits not just for NAS, but for applications of LLMs. Whereas other NAS methods rely on a hand-designed search space consisting of a finite set of building blocks, EvoPrompting can generate potentially any architecture, since it directly generates code instead of discrete building blocks. This is flexible, more expressive, and easier to use than approaches that require the hand-designed search space. The generated architecture may include components that the human designer did not think to include in the search space.
> - We additionally show in our ablation studies that EvoPrompting is significantly more effective than applying a language model alone – it far more effectively completes this difficult task that LLMs had not been evaluated on before and even proposes novel architecture designs that beat the current SoTA without adding more model capacity. Our search requires only 1600 samples for both MNIST-1D and CLRS, while both **finding SoTA architectures and improving the Pareto frontier of model size/test accuracy**. Previous multi-trial NAS efforts have required anywhere from O(10K) ([So et al.](https://arxiv.org/abs/2109.08668)) to O(1T) ([Real et al.](http://proceedings.mlr.press/v119/real20a/real20a.pdf)) examples.
> - While developing novel GNNs may not be quite as competitive as developing more common architectures like Transformers, they are still very important to parts of the ML community. To scale up to more competitive tasks would require more compute than we had the budget for – for example, pre-training competitive language models is very expensive.
>
> "ablation studies are missing for the verification of LLM better than the original crossover operator and mutation operator"
> - There is unfortunately no "original" crossover and mutation operator - these operators vary vastly in design across NAS works because they are search-space specific. One of the benefits of EvoPrompting is that it does not require a hand designed search space and so comparing to such a search space and its corresponding mutator operators would introduce too many confounders to be useful.
>  - We compared implicitly to the "conventional" mutators and crossovers in methods published on the NATS-Bench benchmark.
> - We also conducted ablation studies of simpler crossover/mutation operators that are natural with an LM setup, such as using naive few-shot prompting or selecting random parents to crossover from.
>
> "the fitness function is quite odd because it is obtained through the test from the authors, while the motivation is not clear"
> - Our fitness function is simply a way to weight the val. error by model size, which penalizes the more trivial designs that improve upon validation accuracy simply by increasing the number of parameters. It is common in evolutionary NAS to either use the val. accuracy as the fitness function ([Liu et al.](https://ieeexplore.ieee.org/abstract/document/9508774)) or a weighted combination of the val. accuracy and other factors ([Branke et al.](https://link.springer.com/book/10.1007/978-3-540-88908-3), [Groh and Kist](https://ieeexplore.ieee.org/abstract/document/10189194))
>
> "The main problem is the experiments are not common ones within the community of neural architecture search, and the compared algorithms are not the state of the arts"
> - We did conduct a comparison against several common multi-trial and weight sharing NAS methods (including the commonly used DARTS) on the NATS-Bench benchmark, which is **common and standard within the NAS community and covers the widely known CIFAR-10, CIFAR-100, and ImageNet benchmarks**. This comparison does handicap EvoPrompting in several key ways (discussed in Appendix A.9 and referenced in the conclusion), but EvoPrompting still performs comparably to the other NATS-Bench methods, despite not being able to use its full functionality.
> - However, multiple past works ([Yu et al.](https://arxiv.org/abs/1902.08142), [Bender et al.](https://arxiv.org/abs/2008.06120), [Li and Talwalker](https://arxiv.org/abs/1902.07638)) have noted the unfairness and confounding nature of NAS comparisons with unequal search spaces. Since EvoPrompting is not designed for use with a finite and discrete search space (i.e. a set of pre-defined building blocks) and instead offers a flexible search method capable of generating more novel and varied architectures via directly generating code, there is no clear and fair way to directly compare EvoPrompting against other NAS methods.

---

> > ### Author Response · Authors · 2023-08-17
> > **Re: ablations**
> >
> > Thanks for reviewing our rebuttal! We appreciate it. Concerning the ablations, we wanted to re-emphasize that:
> >
> > (1) Our paper includes **ablation studies of simpler crossover/mutation operators** that are natural with an LM setup, such as using naive few-shot prompting or selecting random parents to crossover from.
> >
> > (2) There is unfortunately no "original" crossover and mutation operator - these operators vary vastly in design across NAS works because they are search-space specific. One of the benefits of EvoPrompting is that it does not require a hand designed search space and so comparing to such a search space and its corresponding mutator operators would introduce too many confounders to be useful.
> >
> > (3) We have also implicitly compared against the crossover/mutation operators used in the methods published on the NATS-Bench benchmark.
> >
> > **An important question**: Maybe it'd be helpful if you could specify what you mean by the "original crossover operator and mutation operator"? It's hard for us to conduct these experiments or respond without understanding clearly what this experiment would be. Thank you!

---

### Official Review · Reviewer_xBLj · 2023-07-03

**Soundness:** 3 good
**Presentation:** 4 excellent
**Contribution:** 2 fair
**Rating:** 4
**Confidence:** 4

**Summary:**

The paper presented an evolutional neural architecture search method that utilizing LLM as mutator. During the evolutional search, the LLM is updated according to the evaluation result. Meanwhile the evaluation results are fed back to the LLM during the mutation process in the format of in-context learning.

**Strengths:**

The work explores a novel direction of using LLMs in NAS. Many efforts have been made to design this approach to enable the LLM in NAS as it is not particularly trained on this task. The experimental results also proved the naive way of using LLM is not as effective as the proposed one.

**Weaknesses:**

One missing part of the paper is why we will be interested in using LLM for NAS. Also the experiment mainly demonstrates the proposed method is better than naive prompting methods. How does the NAS with LLM compare  to the existing NAS method?


The experiments are better to be conducted with large scale realistic datasets.

**Questions:**

1. How does the proposed method perform comparing to the existing non-LLM method? Given the same number of search? And what’s the computation resource used for LLM search method and nonLLM methods?

2. The author mentioned the CNN architecture search might be in the training set of the LLM. Does author train the LLM? Or is it a publicly accessible model?

**Limitations:**

The authors have not discussed the limitation and border impact. However, the reviewer has not seen any significant limitation of the method, and potential negative border impact.

---

> ### Author Rebuttal · Authors · 2023-08-10
>
> We thank the reviewer for the time and effort spent on providing valuable feedback on our work. Below we provide responses to the concerns mentioned.
>
> "One missing part of the paper is why we will be interested in using LLM for NAS"
> - Lines 139-150 in our paper are relevant to this question. LLMs have demonstrated exceptional competence at generating code ([Xu et al.](https://arxiv.org/abs/2202.13169)), and since all neural architectures can be encoded via code, LLMs offer a more expressive search tool than the finite hand-designed search spaces that most NAS algorithms entail. The LLM may even generate components that are not typically seen in NAS search spaces. Furthermore, using an LLM introduces other skills that the LLM may have learned from its training, such as the ability to generate high-likelihood outputs, to condition on a target reward, and to transfer learnings from other skills seen in the data. LLMs also introduce less manual engineering/human bias than hand-designing an optimal search space would be. Lastly, LLMs can also be tuned to better adapt to the task, making the LLM an adaptive crossover operator.
>
> "How does the NAS with LLM compare to the existing NAS method?"
> - We did attempt a comparison against several common multi-trial and weight sharing NAS methods on the NATS-Bench benchmark. This comparison does handicap EvoPrompting in several key ways (discussed in App. A.9 and referenced in the conclusion), but EvoPrompting still performs comparably to the other NATS-Bench methods, despite not being able to use its full functionality.
> - However, multiple past works ([Yu et al.](https://arxiv.org/abs/1902.08142), [Bender et al.](https://arxiv.org/abs/2008.06120), [Li and Talwalker](https://arxiv.org/abs/1902.07638)) have noted the unfairness and confounding nature of NAS comparisons with unequal search spaces. Since EvoPrompting is not designed for use with a finite and discrete search space (i.e. a set of pre-defined building blocks) and instead offers a flexible search method capable of generating more novel and varied architectures via directly generating code, there is no clear and fair way to directly compare EvoPrompting against other NAS methods.
>
> "The experiments are better to be conducted with large scale realistic datasets."
> - We were unfortunately compute-constrained – you may notice that we only used a single P100 GPU to train each child model and only prompt-tuned (instead of fine-tuning) the LM. We agree that some interesting results could be obtained from evaluating EvoPrompting on designing larger scale architectures, but given that EvoPrompting was already able to propose multiple novel, non-trivial, and state-of-the-art architectures for a difficult algorithmic reasoning task, we believe this demonstrates the promise and broader applicability of EvoPrompting. Furthermore, our approach often accomplished this via designing architectures of smaller size than the state-of-the-art, suggesting that EvoPrompting can design architectures that scale more efficiently.
>
> "The author mentioned the CNN architecture search might be in the training set of the LLM."
> - Sorry for the confusion – we do not mean that the exact architectures generated were in the training set. We only mean that since LLMs are trained on large corpora scraped from the Internet, it has likely seen code for many different convolutional nets before. However, the experiments on MNIST1D are still interesting because it is non-trivial to optimize the best form of CNN needed for a particular task and EvoPrompting was not limited to proposing CNNs. Since EvoPrompting accomplished this task significantly better and more efficiently than the baseline methods, this section shows the promise of EvoPrompting for optimizing neural architectures. But to further address this issue, we also included the CLRS experiments, since this is a newer benchmark and GNNs are studied much less. The novel state-of-the-art GNNs suggested by EvoPrompting contained modifications that none of the authors had seen in prior work, and therefore seemed much less likely to be purely a result of copying the training data.
>
> "Does author train the LLM? Or is it a publicly accessible model?"
> - We did not pretrain or finetune the LLM ourselves – we only prompt-tuned it as part of our algorithm (Alg. 1, step 11). It is a publicly available model, but we elided the model name for anonymity reasons. Thanks for pointing out the ambiguity, we will be sure to clarify these details in the next version of the paper.
>
> "How does the proposed method perform comparing to the existing non-LLM method? Given the same number of search? And what’s the computation resource used for LLM search method and nonLLM methods?"
> - We presented a comparison against non-LLM NAS methods in App. A.9 on NATS-Bench. However, we noted that these results are difficult to interpret due to the unequal search spaces. When handicapped in this way, EvoPrompting performs comparably to the other NAS methods. (However, we believe that one of EvoPrompting's key strengths is not represented in this comparison – i.e. its ability to generate architecture components that are not represented in traditional search spaces.) We followed the time budgets recommended in [Dong et al.](https://arxiv.org/pdf/2009.00437.pdf) and kept the other hyperparams the same, meaning that EvoPrompting always completed its search with fewer than 1600 samples, whereas most methods involve anywhere from O(10K) ([So et al.](https://arxiv.org/abs/2109.08668)) to O(1T) ([Real et al.](http://proceedings.mlr.press/v119/real20a/real20a.pdf)) samples. However, the NATS-Bench paper did not indicate the no. of samples, FLOPS, or GPU-hours required by each method.
> - We used ~130 and ~800 GPU hours to train the child models in the MNIST-1D and CLRS experiments, respectively. (However, this cost would likely be shared by other multi-trial NAS methods.) We used ~24 TPU hours on LM inference and prompt-tuning.

---

> > ### Comment · Reviewer_xBLj · 2023-08-17
> > **Thanks for the response, my score is unchanged.**
> >
> > Thanks authors for the responses. I generally agree with most the intuition and potential benefits of LLMs that authors mentioned in the response: flexible space, knowledge transferred from other training tasks that could lead to high probability good results, etc. However, I don't think the experimental results solidly supports those benefits. Especially, A.9 is emphasized but its result can not conclude any of the baselines or the EvoPrompt is better than others.

---

> > > ### Author Response · Authors · 2023-08-19
> > > **NATS-Bench results**
> > >
> > > Thanks for responding to our rebuttal, we appreciate it!
> > >
> > > - Re: NATS-Bench results - we noted in both App. A.9 and our previous response that this comparison is unfair and handicaps EvoPrompting in multiple ways, since it was not designed for this kind of use. (We also evaluated our method with far fewer samples, the details of which are mentioned in the previous response.) Furthermore, multiple past works ([Yu et al.](https://arxiv.org/abs/1902.08142), [Bender et al.](https://arxiv.org/abs/2008.06120), [Li and Talwalker](https://arxiv.org/abs/1902.07638)) have also highlighted the confounding nature and unfairness of comparing NAS techniques across unequal search spaces.
> > >
> > > - We don't claim that EvoPrompting is strictly better than the other methods evaluated in [Dong et al.](https://arxiv.org/pdf/2009.00437.pdf) (for this particular setting), only that it performs *comparably* in this setting, even despite the handicapping, the smaller amount of trials, and the unequal search spaces. (On every NATS-Bench task considered, EvoPrompting has val/test accuracy that is in the middle of the pack.) EvoPrompting is fundamentally a **different approach designed to support a wider variety of settings** (e.g. settings without hand-designed, finite search spaces) than the methods compared to in the NATS-Bench evaluation.
> > >
> > > - EvoPrompting is better suited for settings without a pre-designed search space, as we demonstrate with our CLRS experiments -- our approach is able to propose **novel and non-trivial SoTA architectures** that generalize to other algorithms not seen during the search itself. (Furthermore, none of the ablated methods or baselines we considered accomplished this.) The previous SoTA (Triplet-GMPNN) was designed with careful and thoughtful manual design and experimentation. It is meaningful that our approach can propose an architecture that significantly out-performs Triplet-GMPNN, even using very few samples and without a hand-designed search space.

---

### Official Review · Reviewer_HTgv · 2023-07-06

**Soundness:** 2 fair
**Presentation:** 2 fair
**Contribution:** 2 fair
**Rating:** 7
**Confidence:** 4

**Summary:**

This is an example of LLMs being used in evolutionary algorithms. The authors use an LLM to crossover code snippets that execute to define a graph or neural architecture. It does this by generative means, not syntactic manipulation as typically in EAs. They use code in the prompt for context and they improve the crossover by including code snippets that resulted in better results. The prompt also has ranges for the desired size of the model and accuracy what are set relative to a parent model, allowing them to guide an incremental improvement in accuracy while keeping model size in check.   They empirically evaluate on MNIST-1D and on CLRS. On MNIST-1D they get variants in terms of accuracy and model size that are comparable to human designs or naive few shot prompting. On CLRS they get novel architectures that are better than a rather modest benchmark on 21/30 tasks (related to algorithm design).  The added bonus is a method that improves prompts through evolutionary search.

**Strengths:**

EvoPrompting is based on in-context prompting and the variation of the "candidate solution" occurs on a code representation.  This changes the search space and makes problem solving more flexible and less reliant on adhoc choices on a different representation level.

 In general, the approach is applicable to LM tasks that rely on in-context learning (ICL) or prompt-tuning.  These approaches don't need gradients.

The results are promising on the CLRS Algorithmic Reasoning Benchmark but not compared to other NAS techniques.  Novel architectures were discovered and they were smaller and had better accuracy than the benchmark.


**Weaknesses:**

The choice of what's in the appendix and in the paper must be hard but I feel at least one code snippet should be shown.  The paper could do with one entire example of the prompt and response.

The search space variation in the MNIST-1D benchmark was modest. For the comparisons with CLRS, one architecture was the compared benchmark for all programs, while EvoPrompting was judged when it was allowed to evolve a a solution for each problem independently.  Authors mention how one evolved model was evaluated with 3 different problems but no others.

There is no comparison of the effort expanded to evolve and test vs NAS by another means.

If the generality of the approach as stated by the authors is to be taken seriously, evidence beyond the narrow scope of code variation and NAS  is necessary.  There are many many more problems than MNIST-1D NAS or CLRS NAS.


**Questions:**

How would evoprompting do on the problems in the GP symbolic regression benchmarks? How do they differ from CLRS and why or why are they not applicable here?
Can you compare to NAS?
How would you select one model from the 21 or 31 options and train them to be tested on all the other problems?

What if you changed the seed architectures? They seem pretty good in both problems.

---

> ### Author Rebuttal · Authors · 2023-08-10
>
> Thank you for your thoughtful and thorough feedback - we appreciate the significant time and effort this took. Your questions also gave us very interesting thoughts and directions to think about. We respond to some of the feedback and questions below:
>
> "...not compared to other NAS techniques...Can you compare to NAS?"
> - We actually did conduct a limited comparison of EvoPrompting to other common multi-trial and weight sharing NAS methods on the NATS-Bench benchmark. This comparison does handicap EvoPrompting in several key ways (discussed in Appendix A.9 and referenced in the conclusion), but EvoPrompting still performs comparably to the other NATS-Bench methods, despite not being able to use its full functionality.
> - However, multiple past works ([Yu et al.](https://arxiv.org/abs/1902.08142), [Bender et al.](https://arxiv.org/abs/2008.06120), [Li and Talwalker](https://arxiv.org/abs/1902.07638)) have noted the unfairness and confounding nature of NAS comparisons with unequal search spaces. Since EvoPrompting is not designed for use with a finite and discrete search space (i.e. a set of pre-defined building blocks) and instead offers a flexible search method capable of generating more novel and varied architectures via directly generating code, there is no clear and fair way to directly compare EvoPrompting against other NAS methods.
>
> "The choice of what's in the appendix and in the paper must be hard but I feel at least one code snippet should be shown. The paper could do with one entire example of the prompt and response."
> - This is a good point – having an example would be helpful for the reader to understand both the context of how EvoPrompting works, and emphasize how the LLM is able to condition upon the desired test accuracy and model size. We'll be sure to move an example from the appendix into the paper in the next version.
>
> "For the comparisons with CLRS, one architecture was the compared benchmark for all programs, while EvoPrompting was judged when it was allowed to evolve a a solution for each problem independently. Authors mention how one evolved model was evaluated with 3 different problems but no others."
> - This is not quite correct – when applying EvoPrompting on the CLRS benchmark, we only applied evolution using validation metrics from a single algorithmic task at a time. However, at the end of EvoPrompting we then trained and evaluated the most fit model separately on all 30 algorithmic tasks in the CLRS benchmark to demonstrate how the suggested design could generalize to other algorithmic tasks that were not seen during EvoPrompting. We applied this process to a total of 3 tasks in the CLRS benchmark due to both computational constraints and there being more headroom in those tasks than others.
> Similarly, the previous state-of-the-art architecture that we compared to (Triplet-GMPNN from [Ibarz et al.](https://arxiv.org/pdf/2209.11142.pdf)) was trained and evaluated separately on the 30 tasks. However, the authors hand-designed this algorithm, so it is not clear how many of the tasks were seen and used as validation during the design process.
>
> "The search space variation in the MNIST-1D benchmark was modest."
> - The search space itself in the MNIST-1D experiments was expansive – the LLM could generate any architecture (that could be expressed with code of length less than its maximum output length), and indeed many of the candidate designs included modifications beyond variations of CNNs, such as attention modules, recurrent or fully-connected layers, etc. However, the best performing child models were mostly optimized versions of CNNs.
> - The CLRS experiments helped address the limitations of the MNIST-1D experiments - the resulting models were far more varied and there was significantly less alignment of the suggested designs with "standard" modules such as attention, recurrence, etc.
>
> "There is no comparison of the effort expanded to evolve and test vs NAS by another means."
> - We mentioned in section 4.1 that open-ended multi-trial NAS methods often require on the order of  anywhere from O(10K) ([So et al.](https://arxiv.org/abs/2109.08668)) to O(1T) ([Real et al.](http://proceedings.mlr.press/v119/real20a/real20a.pdf)) samples, whereas our experiments only required ~1600.
> - In our NATS-Bench comparison (referenced in the conclusion but detailed in Appendix A.9), we used the time budget recommended by [Dong et al.](https://arxiv.org/pdf/2009.00437.pdf) to demonstrate how EvoPrompting still performed comparably to other NAS methods under the same time budgets. However, the NATS-Bench paper does not mention other details about the exact computational resources allocated to each technique, which makes comparison difficult.
> - Nevertheless, we're happy to include more details about the exact computational resources we used in our experiments – we currently mention that each child model was trained on a single P100 (not necessarily to convergence). We will insert more detailed estimates of the total amount of compute (~130 GPU hours and ~800 GPU hours for training the MNIST-1D and CLRS child models, respectively, which is a cost that would likely be shared by other multi-trial NAS methods; and ~24 TPU hours for running inference and prompt-tuning on the LLM across both tasks).
>
> "How would evoprompting do on the problems in the GP symbolic regression benchmarks?"
> - Given that the symbolic regression benchmarks have a continuous fitness which allows for iteratively improving the proposed answer, we would expect EvoPrompting to work reasonably well.
>
> "What if you changed the seed architectures? They seem pretty good in both problems."
> - We used a handful of baseline models (mostly suggested in prior literature) as the seeds for our analyses, in the hopes of finding architectures which would improve over what has already been suggested in past work. While bad starting models would likely damage performance, the process would still improve performance over the seeds.

---

> > ### Comment · Reviewer_HTgv · 2023-08-19
> >
> > Thank you for your clarifications, they reinforce my evaluation.  The comment about search space variation in the MNIST-1D benchmark being modest was about the range of variation, not the search space.

---

### Official Review · Reviewer_VcYp · 2023-07-07

**Soundness:** 4 excellent
**Presentation:** 4 excellent
**Contribution:** 3 good
**Rating:** 7
**Confidence:** 4

**Summary:**

This paper introduces a method that uses LLMs as mutation and crossover operators in an evolutionary search process that generates diverse and high performing neural architectures. They evaluate their method on two datasets, MNIST-1D and CLRS, a benchmark measuring algorithmic reasoning. The evoprompting process generated SOTA models on CLRS and performed well on MNIST-1D. Importantly it generated nontrivial architectures.

**Strengths:**

Originality:
The concept of using a sequence model to generate neural network architectures in NAS is not new (Neural Architecture Search with Reinforcement Learning by Zoph), but was previously very constrained because the parameters the sequence model output were simple, things like width and kernel size for a CNN. This model is much more expressive, can describe a much broader set of architectures, because it generates code. Combining LLMs and evolution is also not strictly novel, see Language Model Crossover: Variation through Few-Shot Prompting, but I believe this is quite different.
Quality:
I think the quality of the writing and the research is high. They show state of the art on an admittedly new benchmark. Nobody has done this before, they show that the sample complexity of the search is good.
Significance:
It demonstrates a practical way to leverage the capabilities of language models for complex tasks which has wide ranging implications. I could imagine this being extended to program synthesis for example.

I like that the authors included some sample architectures in the supplemental materials. It looks like the search discovered nontrivial architectures.


**Weaknesses:**

The paper could be improved by:
- Comparing to other NAS approaches
- Evaluating on larger tasks. I don't fault the authors at all for this, NAS is expensive and I like that they used MNIST-1D, but more domains would strengthen the claims.

**Questions:**

How do you think this would fare with larger models?
Are there any specific domains where you think this approach would struggle?

**Limitations:**

Yes

---

> ### Author Rebuttal · Authors · 2023-08-10
>
> Thank you for your detailed and insightful comments – we particularly appreciated your references to other related work and how those relate or are different from our work. We detail below our responses to some of the weaknesses and questions.
>
> "Weaknesses: Comparing to other NAS approaches"
>
> - We presented a limited comparison of EvoPrompting to other common multi-trial and weight sharing NAS methods on the NATS-Bench benchmark, which is common and standard within the NAS community. This comparison does handicap EvoPrompting in several key ways (discussed in Appendix A.9 and referenced in the conclusion), but EvoPrompting still performs comparably to the other NATS-Bench methods, despite not being able to use its full functionality.
> - However, multiple past works ([Yu et al.](https://arxiv.org/abs/1902.08142), [Bender et al.](https://arxiv.org/abs/2008.06120), [Li and Talwalker](https://arxiv.org/abs/1902.07638)) have noted the unfairness and confounding nature of NAS comparisons with unequal search spaces. Since EvoPrompting is not designed for use with a finite and discrete search space (i.e. a set of pre-defined building blocks) and instead offers a flexible search method capable of generating more novel and varied architectures via directly generating code, there is no clear and fair way to directly compare EvoPrompting against other NAS methods.
>
> "How do you think this would fare with larger models?"
> - Although it's hard to know without further extensive experiments that we cannot conduct with our current computational constraints, we suspect that using EvoPrompting to improve larger architectures (e.g. Transformers) would require a more modular approach, similar to our CLRS experiments. That is, EvoPrompting could be applied to a single module at once (e.g. the attention mechanism) or applied at a higher level (i.e. being prompted to design the optimal combination of modules, given a list of the already-designed modules). Due to the inherent flexibility of LLMs, we anticipate that there are a wide variety of ways that EvoPrompting could be used to tackle this problem. The use of a longer LLM context would also be important, but the current research on how best to adapt LLMs to effectively use a longer context is still burgeoning. We look forward to exploring this in follow-up work.
>
> "Combining LLMs and evolution is also not strictly novel, see Language Model Crossover: Variation through Few-Shot Prompting, but I believe this is quite different."
> - This is true, and a great point – we compare and contrast our work with this work in the Related Works section. We also note that this is concurrent work that was released at a very similar time as our paper.
>
> "Are there any specific domains where you think this approach would struggle?"
> - Our approach currently relies on the existence of a fitness gradient – if the rewards are more sparse, we would expect EvoPrompting to struggle to find better solutions. This is a general weakness of many evolutionary or RL-based approaches. However, if iterative improvement can consistently improve reward, then we expect EvoPrompting to be able to find high fitness solutions.

---

### Official Review · Reviewer_fsDm · 2023-07-07

**Soundness:** 3 good
**Presentation:** 4 excellent
**Contribution:** 4 excellent
**Rating:** 7
**Confidence:** 3

**Summary:**

The paper explores prompt tuning for neural architecture search. While language models can produce code snippets with prompting, it is often quite difficult for language models to succeed at this task. The authors propose EvoPrompting, a prompt tuning method that produces neural architectures. The authors experiment with MNIST-1D where they find that EvoPrompting produces better performing neural networks with fewer parameters than the baselines. They then test their method on producing graph neural networks for algorithmic reasoning and find that they achieve the state-of-the-art on 21 out of 30 tasks.

**Strengths:**

This is a high quality work with strong results on the CLRS algorithmic benchmark. The authors first carefully validate their method on a simple task MNIST-1D and find that their method produces smaller but effective neural networks compared to existing baselines. The results on the CLRS benchmark are quite impressive, sometimes improving higher than 20% (QuickSort) over the baseline while still having the same number of parameters.



**Weaknesses:**

Analysis of the language model: the paper uses a large language model as a blackbox but fails to provide analysis about the language model itself. The paper trains a new 62B parameter decoder-only language model trained on conversational, web, and code documents. It would be useful to know if there are equivalent open-source models such as StarCoder that can be used to replicate the results? Furthermore, it is unclear how important the model size will affect the capabilities. It would be interesting to the readers to know the performance on CLRS across model sizes if possible.

Unclear use of “soft prompt-tuning”: The work uses soft prompting in each round to find better in-context learning prompts. However, it is not clear if soft prompting is the best method to find the relevant prompts. It would be helpful to compare soft prompting with related techniques such as adapters [a], LoRA [b], and T-Few [c].

Nit: the work could provide more background regarding genetic programming similar to [d].

[a] Parameter-Efficient Transfer Learning for NLP. ICML 2020.

[b] LoRA: Low-Rank Adaptation of Large Language Models. ICLR 2022.

[c] Improving In-Context Few-Shot Learning via Self-Supervised Training. NeurIPS 2022.

[d] Evolution through Large Models. 2022.


**Questions:**

See the questions for weaknesses.

---

> ### Author Rebuttal · Authors · 2023-08-10
>
> The authors thank the reviewer for taking the time to carefully note both the strengths and weaknesses of our approach – we particularly appreciated the detailed notes about the particulars of applying LLMs and the associated implications for the performance of EvoPrompting.
>
> "the paper uses a large language model as a blackbox but fails to provide analysis about the language model itself.  The paper trains a new 62B parameter decoder-only language model trained on conversational, web, and code documents."
> - Sorry for the confusion, we did not pre-train or fine-tune the LLM ourselves – it is an off-the-shelf LLM that is currently publicly available. We elided the name for anonymity purposes but will re-insert the details if the paper is accepted. We only prompt-tuned the model as part of the EvoPrompting algorithm.
> - Since this is an off-the-shelf LLM that already has a paper published about its general performance across a variety of both language and code generation tasks, we did not feel the need to additionally analyze this model. We will include a citation to this LLM paper in the next version of the paper.
>
> "Furthermore, it is unclear how important the model size will affect the capabilities. It would be interesting to the readers to know the performance on CLRS across model sizes if possible."
> - We agree that it would be interesting to see how EvoPrompting performance scales as a function of LLM size, but unfortunately did not have the computational constraints to conduct extensive experiments across many LLM sizes.
> - We did try preliminary experiments with a smaller LLM (~8B parameters) and found that EvoPrompting took a longer time and more samples in order to reach similar accuracies as the 64B model. The primary issue was that the smaller LLM was more likely to generate incorrect syntax or repetitive programs, thereby requiring sampling more programs than the 64B model required. This is a common issue for LLMs with less capacity, but it is possible that pre-training on more data could help resolve this. We did not have enough resources to pre-train an LLM ourselves, but hope to explore this with other LLMs in follow-up work.
>
> "Unclear use of “soft prompt-tuning”: The work uses soft prompting in each round to find better in-context learning prompts. However, it is not clear if soft prompting is the best method to find the relevant prompts"
> - Our choice of soft prompt-tuning (instead of prefix-tuning, adapter-based methods, or LoRA), was based both on ease of implementation and parameter efficiency. Our soft prompts had dimension 16. However, prefix-tuning ([Li and Liang](https://arxiv.org/abs/2101.00190)) requires adding a prefix to every hidden layer of the model, usually resulting in the tuning and storage of 0.1% of the number of model parameters, which would be 64M parameters in our case. Similarly, adapters ([Houlsby et al.](https://arxiv.org/abs/1902.00751)) usually require tuning and storage of 2-4% of the number of model parameters (~1.28B-2.56B parameters in our case). The dimensionality of LoRA updates vary depending on the desired rank, but the smallest number of trainable parameters explored in [Hu et al.](https://arxiv.org/abs/2106.09685) is 0.77M parameters.
> - Given the computational resources, other parameter-efficient methods like LoRA, prefix-tuning, or adapters could result in even better performance, but EvoPrompting can use any model tuning method that involves likelihood maximization.
> - We sadly did not have enough compute to run a thorough comparison of all these techniques, but we think it is still promising that our technique works with only a soft prompt of dimension 16 – we cannot claim that soft prompt-tuning is strictly the most optimal way to adapt the LLM during each round of EvoPrompting, but it requires tuning the fewest parameters. Given how expensive NAS methods already are, we think this is an important point.

---

> > ### Comment · Reviewer_fsDm · 2023-08-16
> > **Acknowledgement**
> >
> > Thank you for your response. The authors have clarified my questions. I will keep my score at 7.

---

### Comment · Area_Chair_L8yk · 2023-08-19

Dear Reviewers,

The authors and I are eager to ascertain whether the author responses have effectively addressed your concerns. Due to the limited time allocated for the author-review discussion phase, we strongly encourage you to provide your direct feedback to the authors.

Thanks for your hard work.

Best regards,

AC

---

### Decision · Program_Chairs · 2023-09-21

**Decision:**

Accept (poster)

**Comment:**

This paper propose a quite interesting and innovative idea to perform neural architecture search through code generation LLM which can play as a general adaptive mutation and cross over operators. This way is more expressive and intuitive to encode the architecture through its implementation code and let code LLM learn to select the best fits. It also shows strong empirical results compared to baseline of conventional NAS approaches. The reviewers have concerns that the application task on benchmark are limited and the strong results might not generalize to more complex tasks. The authors are encouraged to take into account the feedback and input provided by the reviewers to improve this work. Overall, the merits outweigh the drawbacks, I recommend accepting this submission.